



# Mountain-wave induced polar stratospheric clouds and their representation in the global chemistry model ICON-ART

Michael Weimer[1,2], Jennifer Buchmüller[2,*], Lars Hoffmann[3], Ole Kirner[1], Beiping Luo[4],
Roland Ruhnke[2], Michael Steiner[5], Ines Tritscher[6], and Peter Braesicke[2]

[1]Steinbuch Centre for Computing, Karlsruhe Institute of Technology (KIT), Eggenstein-Leopoldshafen, Germany
[2]Institute of Meteorology and Climate Research, Karlsruhe Institute of Technology (KIT), Eggenstein-Leopoldshafen, Germany
[3]Jülich Supercomputing Centre, Forschungszentrum Jülich, Jülich, Germany
[4]Institute for Atmospheric and Climate Science, ETH Zurich, Switzerland
[5]Laboratory for Air Pollution / Environmental Technology, EMPA, Switzerland
[6]Institute of Energy and Climate Research: Stratosphere (IEK-7), Forschungszentrum Jülich, Jülich, Germany
[*]now at: Steinbuch Centre for Computing, Karlsruhe Institute of Technology (KIT), Eggenstein-Leopoldshafen, Germany

**Correspondence:** Michael Weimer (michael.weimer@kit.edu)

**Abstract.** Polar stratospheric clouds (PSCs) are a driver for ozone depletion in the lower polar stratosphere. They provide surfaces for heterogeneous reactions activating chlorine and bromine reservoir species during the polar night. PSCs are represented in current global chemistry-climate models, but one process is still a challenge: the representation of PSCs formed locally in conjunction with unresolved mountain waves. In this study, we present simulations with the ICOsahedral Nonhydrostatic mod-

elling framework (ICON) with its extension for Aerosols and Reactive Trace gases (ART) that include local grid refinements (nesting) with two-way interaction. Here, the nesting is set up around the Antarctic Peninsula which is a well-known hot spot for the generation of mountain waves in the southern hemisphere. We compare our model results with satellite measurements from the Cloud-Aerosol LIdar with Orthogonal Polarisation (CALIOP) and the Atmospheric InfraRed Sounder (AIRS). We study a mountain wave event that took place from 19 to 29 July 2008 and find similar structures of PSCs as well as a fairly real-

istic development of the mountain wave in the Antarctic Peninsula nest. We compare a global simulation without nesting with the nested configuration to show the benefit. Although the mountain waves cannot be resolved adequately in the used global resolution (about $160\,\text{km}$), their effect from the nested regions (about $80$ and $40\,\text{km}$) on the global domain is represented. Thus, we show in this study that by using the two-way nesting technique the gap between directly resolved mountain-wave induced PSCs and their representation and effect on chemistry in coarse global resolutions can be bridged by the ICON-ART model.

## 1  Introduction

Polar stratospheric clouds (PSCs) play a key role in explaining the ozone loss in the polar stratosphere during local spring (e.g., Solomon et al., 1986; Solomon, 1999; Braesicke et al., 2018). Heterogeneous reactions on the surface of PSCs lead to activation of chlorine and bromine species during the polar night, thus enhancing the catalytic ozone depletion cycles as soon as the sun rises (e.g., Solomon, 1999). In addition, PSCs can irreversibly remove nitrogen-containing species by sedimentation,





today known as denitrification, thus extending the period of low ozone concentrations during polar spring (e.g., Waibel et al., 1999).

One of the processes under investigation is PSCs formed in conjunction with mountain waves. It is known that mountain-wave induced PSCs have a significant influence on the ozone depletion over Antarctica and Arctic (e.g., McDonald et al., 2009; Alexander et al., 2011). However, the simulation of mountain-wave induced PSCs with atmospheric chemistry models (ACMs)
is still a challenge (e.g., Orr et al., 2015). With horizontal resolutions in the order of a few hundreds of kilometres, ACMs are not able to resolve the orography adequately to directly simulate mountain waves (Morgenstern et al., 2010). Thus, mountain waves and mountain-wave induced PSCs either have to be parametrised (Orr et al., 2015; Zhu et al., 2017; Orr et al., 2020) or have to be calculated as post-processing via Lagrangian models (e.g., Mann et al., 2005) or higher resolved mesoscale models (e.g., Fueglistaler et al., 2003; Eckermann et al., 2006), which have to be limited to a specific region on Earth as discussed e.g.
in Weimer et al. (2016). A method to bridge this gap for interactive calculations is missing so far.

Three types of PSCs are known to exist: (1) solid nitric acid trihydrate (NAT) particles, mostly responsible for denitrification (e.g., Fahey et al., 2001), (2) liquid supercooled ternary solution (STS) droplets which grow from the background sulfate aerosol (Junge et al., 1961; Wilka et al., 2018) by taking up nitric acid ($HNO_3$) and on which the major fraction of chlorine and bromine activation takes place (e.g., Peter, 1997; Kirner et al., 2015) and (3) ice clouds forming at the lowest temperatures
and leading to dehydration of the lower stratosphere (e.g., Kelly et al., 1989; Khaykin et al., 2013).

Various climatologies of PSCs based on satellite measurements were published in the last years (Spang et al., 2018; Höpfner et al., 2018; Pitts et al., 2018). The application of machine learning for the detection of PSCs from satellite measurements has been discussed recently (Sedona et al., 2020).

Nucleation processes especially of NAT and ice PSCs are still under debate (e.g., Voigt et al., 2018). The question if equilib-
rium or non-equilibrium processes are needed to describe the growth of STS particles is an issue of research (Zhu et al., 2015). In addition, it is also known that mountain waves can induce PSCs which influence the ozone chemistry (e.g., Hoffmann et al., 2017).

Mountain waves (orographic gravity waves) are stationary waves in the lee of a mountain which can develop in a stably stratified atmosphere (e.g., Fritts and Alexander, 2003). They can propagate upwards into the stratosphere and higher (Wright
et al., 2017) and perturb the synoptic temperature field with a local amplitude in the order of up to $\pm 15\,\mathrm{K}$ (Meilinger et al., 1995; Carslaw et al., 1998; Eckermann et al., 2009).

Apart from heterogenenous nucleation of NAT on meteoric dust (e.g., Hoyle et al., 2013; Tritscher et al., 2019), denitrification in the Arctic region is closely connected to mountain wave activity because large NAT particles can be formed in the low temperatures associated with mountain waves in the northern hemisphere (e.g., Tabazadeh et al., 2000). On the other hand,
mountain-wave induced PSCs are also important for the southern hemisphere. Antarctic mountain-wave induced NAT PSCs were detected by Höpfner et al. (2006) for the first time. McDonald et al. (2009) stated that up to $40\,\%$ of PSC formation comes from mountain waves in the early Antarctic winter when temperature is close to that of NAT formation. Alexander et al. (2011) concluded that about $30\,\%$ of all southern hemispheric PSCs can be related to mountain waves.





In this study, mountain-wave induced PSCs are simulated seamlessly with the ICOsahedral Nonhydrostatic modelling frame-
work (ICON, Zängl et al., 2015) and its extension for Aerosols and Reactive Trace gases (ART, Rieger et al., 2015; Weimer
et al., 2017; Schröter et al., 2018). ICON-ART provides the possibility of local grid refinement with two-way interaction so
that a coarse global resolution can be complemented by a region with a refined grid (Reinert et al., 2019), such as the Antarc-
tic Peninsula which is one of the hot spots for the generation of southern hemispheric mountain waves (Bacmeister, 1993;
Bacmeister et al., 1994; McDonald et al., 2009; Alexander et al., 2011; Hoffmann et al., 2017). Therefore, mountain-wave
induced PSCs can be directly simulated with ICON-ART in the refined grid (nest) and their effect can be treated at the global
resolution in the same simulation where they cannot be resolved without the nest.

The ICON-ART model is described in Sect. 2, followed by a description of the simulation in Sect. 3 and the satellite data
used for comparison with ICON-ART in Sect. 4. The model results are compared with CALIOP and AIRS measurements and
mountain-wave induced PSCs are investigated for a typical mountain wave event in Sect. 5. Finally, conclusions and outlook
follow in Sect. 6.

## 2 The ICON-ART model

The ICON model is the operational model for numerical weather prediction at the German Weather Service (DWD, Zängl
et al., 2015). In addition, it can be applied to large eddy simulations (Dipankar et al., 2015) and fully coupled with an ocean
and a land surface model for climate integrations with the climate physics configuration (Giorgetta et al., 2018). The ART
extension has been developed to incorporate aerosols and the atmospheric chemistry into ICON. It can be coupled to ICON in
configurations for numerical weather prediction (Rieger et al., 2015) and allows flexible configurations for weather and climate
integrations (Schröter et al., 2018).

In this study, the chemistry in ICON-ART is based on the Module Efficiently Calculating the Chemistry of the Atmosphere
(MECCA, Sander et al., 2011a) which uses the Kinetic PreProcessor (KPP) to generate Fortran files for solving the specified
chemical mechanism (Sandu and Sander, 2006). Photolysis rates are calculated with the CloudJ module (Prather, 2015; Weimer
et al., 2017). In this study, a system of 142 chemical reactions including 38 photolytic reactions and 11 heterogeneous reactions
on the surface of PSCs is used which is similar to other studies (e.g., Stone et al., 2019; Zambri et al., 2019; Nakajima et al.,
2020) and can be found in Appendix A. Trace gas emissions at the Earth's surface are included by a module described in
Weimer et al. (2017).

The model equations of ICON-ART are discretised horizontally on an icosahedral-triangular C grid (e.g., Staniforth and
Thuburn, 2012; Zängl et al., 2015). The global resolution can be refined by root divisions and bisections of the original icosa-
hedron, resulting in the horizontal resolution description R$n$B$k$, as defined by e.g. Zängl et al. (2015). Vertical discretisation is
performed on generalised smooth-level coordinates (Leuenberger et al., 2010).

For the purpose of detailed simulations around a specific region, the grid can be refined for the area of interest by further
bisections. Here, the parent domain provides boundary conditions for the nested domain. The simulated values in the nested
domain are interpolated to the parent grid with a relaxation-based method (Reinert et al., 2019). Thus, the two-way nesting





enables the model to simulate e.g. mountain waves directly at hot spots and also to impact the variables at a global resolution that is too coarse to represent these processes.

## 2.1 The PSC scheme

In ICON-ART, the three types of PSCs (ice, NAT and STS) are treated separately. Sensitivity simulations showed that the ICON microphysics for ice clouds, operationally computed up to an altitude of $22.5\,\mathrm{km}$, can be extended to the lower stratosphere up to $30\,\mathrm{km}$ (Weimer, 2019). The hydrometeor microphysics in ICON include several nucleation processes interacting with the other hydrometeors as well as sedimentation of the ice particles (Doms et al., 2011). Thus, both nucleation of ice PSCs and dehydration of the lower stratosphere can be captured by the ICON microphysics. A similar approach is used in the Whole

Atmosphere Community Climate Model (WACCM, Zhu et al., 2015). Ice particle radius ($r_\mathrm{ice}$ in m) and particle number concentration ($N_\mathrm{ice}$ in $\mathrm{m}^{-3}$) correspond to the assumed size distributions in the microphysics (Doms et al., 2011):

$$N_\mathrm{ice}(T) = \min\left(5\exp\left[0.304\left(273.15 - T\right)\right], 250 \times 10^3\right) \tag{1}$$

$$r_\mathrm{ice} = \frac{1}{2}\sqrt[3]{\frac{\rho\, q_\mathrm{ice}}{130\, N_\mathrm{ice}}} \tag{2}$$

where $\rho$ is the air density in $\mathrm{kg\,m}^{-3}$, $q_\mathrm{ice}$ is the mass mixing ratio in $\mathrm{kg\,kg}^{-1}$ of water in ice.

Two parametrisations for the microphysics of NAT particles are integrated in ICON-ART. The thermodynamic NAT parametrisation is diagnostic and therefore computes the number of moles of NAT particles in thermodynamic equilibrium, calculates its sedimentation and evaporates the particles again within the same model time step. The volume mixing ratio of $HNO_3$ condensed in NAT ($X_{HNO_3(\mathrm{NAT})}$ in $\mathrm{mol\,mol}^{-1}$) is calculated on the basis of the difference between vapour pressure of $HNO_3$ ($p_{HNO_3}$ in Pa) and the saturation vapour pressure over NAT ($p_\mathrm{sat,NAT}$ in Pa):

$$X_{HNO_3(\mathrm{NAT})} = \frac{p_{HNO_3} - p_\mathrm{sat,NAT}}{p} \tag{3}$$

   The saturation vapour pressure over NAT is calculated according to Hanson and Mauersberger (1988) and $p$ is the air pressure (in Pa). Particle number concentration ($N_\mathrm{NAT}$) and radius ($r_\mathrm{NAT}$) in the thermodynamic NAT parametrisation are calculated using a threshold in the number concentration of $N_\mathrm{NAT,max} = 2.3 \times 10^{-4}\,\mathrm{cm}^{-3}$ which originates from observations (Fahey et al., 2001). Below this threshold, the radius of the NAT particles is set to $0.1\,\mathrm{\mu m}$. Above this threshold, the particle number

concentration is set to $N_\mathrm{NAT,max}$ and the radius of the particles is increased accordingly. This method has already been used in a similar way for solid particles by Kirner et al. (2011).

   The kinetic NAT parametrisation is a non-equilibrium approach based on prognostic equations for the particle mass (see e.g., Seinfeld and Pandis, 2006, p. 539). Carslaw et al. (2002) applied this approach to NAT particles in a Lagrangian model and van den Broek et al. (2004) extended it to Eulerian models.

In the Lagrangian description by Carslaw et al. (2002), the change in NAT particle radius ($r_{\mathrm{NAT},b}$) is calculated prognostically by a diffusive growth

$$\frac{\mathrm{d}r_{\mathrm{NAT},b}}{\mathrm{d}t} = \frac{G_b}{r_{\mathrm{NAT},b}} \tag{4}$$





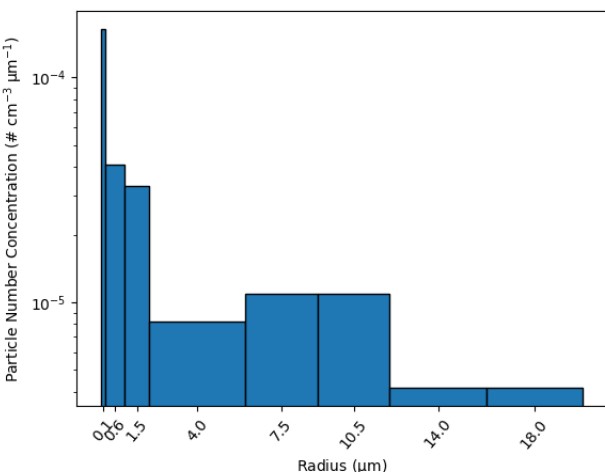

**Figure 1.** Size distribution of NAT particles used in this study. Based on van den Broek et al. (2004).

where $G_b$ (in $\mathrm{m^2\,s^{-1}}$) is a growth factor which depends on the diffusion coefficient of $\mathrm{HNO_3}$ in air ($D_{\mathrm{HNO_3}}$, in $\mathrm{m^2\,s^{-1}}$), the air temperature ($T$, in K) and the saturation difference of the $\mathrm{HNO_3}$ vapour pressure (in Pa):

$$G_b = \frac{D^*_{\mathrm{HNO_3},b} M_{\mathrm{NAT}}}{\rho_{\mathrm{NAT}} R^* T}\left(p_{\mathrm{HNO_3}} - p_{\mathrm{sat,NAT}}\right) \tag{5}$$

with

$$D^*_{\mathrm{HNO_3},b} = \frac{D_{\mathrm{HNO_3}}}{1 + 4\,D_{\mathrm{HNO_3}}/\left(\overline{v}_{\mathrm{HNO_3}}\,r_{\mathrm{NAT},b}\right)} \tag{6}$$

In these equations, $\overline{v}_{\mathrm{HNO_3}}$ is the mean thermal velocity of air molecules (in $\mathrm{m\,s^{-1}}$), $R^*$ stands for the universal constant of an ideal gas in $\mathrm{J\,mol^{-1}\,K^{-1}}$, $\rho_{\mathrm{NAT}}$ is the crystal density of NAT ($1.626 \times 10^6\,\mathrm{g\,m^{-3}}$, Drdla et al., 1993; van den Broek et al., 2004) and $M_{\mathrm{NAT}}$ is the NAT molar mass of $117\,\mathrm{g\,mol^{-1}}$. In one of the Eulerian formulations by van den Broek et al. (2004), this radius change is applied to particles in size bins, but directly converted into a change in the particle number concentration ("FixedRad" approach). This approach is used in this study with a size distribution based on van den Broek et al. (2004) as shown in Fig. 1. That is why we added a $b$ subscript to all variables in Eqs. (4), (5) and (6) that depend on the size bin $b$.

The size distribution of NAT particles can be flexibly specified by the user to investigate its impact on denitrification without any change in the Fortran code but rather changing the respective XML control file (cf. Schröter et al., 2018). Each size bin is defined by radius limits and a maximum particle number concentration, which are kept constant during the simulation. If the calculated particle number concentration in a bin exceeds the maximum, the excess mass is transferred to the next larger size bin. The sum of the maximum particle number concentrations of the bins has to equal the observed value by Fahey et al. (2001), which is $2.3 \times 10^{-4}\,\mathrm{cm^{-3}}$. The size bins are transported as passive tracers in ICON-ART.

Since the calculation of $p_{\mathrm{sat,NAT}}$ by Hanson and Mauersberger (1988) has specific temperature limits, NAT particles are evaporated automatically at temperatures higher than $220\,\mathrm{K}$. For temperatures below $180\,\mathrm{K}$, we calculate $p_{\mathrm{HNO_3}}$ with a constant





temperature of $180\,\mathrm{K}$. These two NAT parametrisations are also implemented in the ECHAM/MESSy Atmospheric Chemistry model (EMAC, Jöckel et al., 2010; Kirner et al., 2011).

Sedimentation of NAT particles, either formed by thermodynamic or kinetic NAT parametrisation, is calculated by a simple
upwind method, using the Stokes velocity of assumed spherical particles (Stokes, 1851).

The microphysics of STS particles in the module are calculated by the scheme first published by Carslaw et al. (1995), with one exception: in the original code by Carslaw et al. (1995) the particle number concentration is set to the constant value of $10\,\mathrm{cm}^{-3}$. We improved this fixed value by applying the mean of all balloon-borne STS measurements by Hervig and Deshler (1998) in order to derive the particle surface area concentration $S_{\mathrm{STS}}$ and the radius $r_{\mathrm{STS}}$ of STS particles from the internally
calculated particle volume concentration $V_{\mathrm{STS}}$:

$$S_{\mathrm{STS}} = 6.068\,(V_{\mathrm{STS}})^{0.671} \tag{7}$$

$$r_{\mathrm{STS}} = \frac{3\,V_{\mathrm{STS}}}{S_{\mathrm{STS}}} \tag{8}$$

In these equations, $V_{\mathrm{STS}}$ has to be given in $\mathrm{\mu m}^3\mathrm{cm}^{-3}$ to get $S_{\mathrm{STS}}$ and $r_{\mathrm{STS}}$ in $\mathrm{\mu m}^2\mathrm{cm}^{-3}$ and $\mathrm{\mu m}$, respectively. Sedimentation is neglected for STS particles since they are too small to result in relevant redistribution of the major constituents $H_2O$, $HNO_3$
and $H_2SO_4$ (Tabazadeh et al., 2000; Considine et al., 2000).

Since the PSCs in the current version of the model do not interact with each other like in a fully coupled PSC scheme (e.g., Zhu et al., 2015), there are essentially two approaches to calculate PSCs: either they are computed with the total (gaseous + liquid + solid) concentrations of $HNO_3$ and $H_2O$ as input for all PSC types (e.g., Kirner et al., 2011) or the PSCs are calculated subsequently with the gaseous fraction that remains after formation of the previously calculated PSC types (operator splitting).
Both approaches have their advantages and disadvantages. We use the second approach because the maximum of $HNO_3$ and $H_2O$ taken up by PSCs cannot exceed the gaseous concentrations in this case. First, ice PSCs are calculated, then NAT PSCs and finally STS PSCs.

Particle radius, particle number concentration and particle surface area concentration are used to calculate the heterogeneous reaction rate constants on the surface of PSCs. For NAT and ice, which can grow to relatively large sizes in the order of tens of
$\mathrm{\mu m}$, the following equation is used to calculate the heterogeneous reaction rate constant, assuming spherical particles (Drdla et al., 1993):

$$k_{\mathrm{het},h,c} = \frac{\gamma_{h,c}\,\pi r_c^2\,\overline{v}_{i(h)}\,N_c}{N_{j(h)}\left(1 + \frac{3\gamma_{h,c}}{4\mathrm{Kn}}\right)}, \quad c \in \{\mathrm{NAT, ice}\} \tag{9}$$

where $\gamma_{h,c}$ is the uptake coefficient (i.e. reaction probability) of heterogeneous reaction $h$ and PSC type $c$, $\overline{v}_{i(h)}$ is the mean thermal velocity of the gaseous reactant $i(h)$ (in $\mathrm{m\,s^{-1}}$), $N_c$ is the PSC number concentration and $N_{j(h)}$ is the number con-
centration of the reactant $j(h)$ adsorbed on the particle (both in $\mathrm{m}^{-3}$). The Knudsen number Kn is calculated by $\lambda_{\mathrm{mfp}}/r_c$ with the mean free path $\lambda_{\mathrm{mfp}}$ of air molecules (in m), calculated according to Kennard (1938). In case of the kinetic NAT parametrisation, Eq. (9) is evaluated for each size bin separately and summed up subsequently.





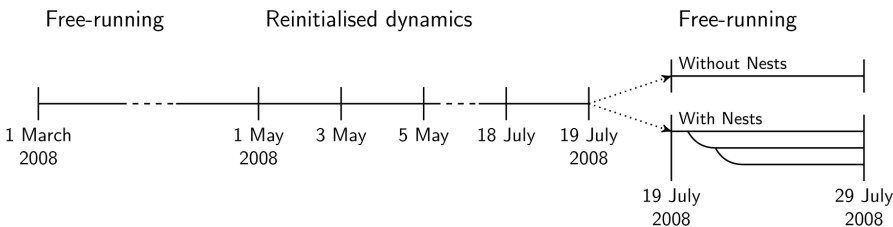

**Figure 2.** Simulation setup in this study: A free-running simulation from 01 March 2008 until 30 April 2008 is followed by a period until 18 July 2008 where the meteorology is reinitialised every second day. During the mountain wave event until 29 July 2008, two simulations are performed: one global simulation without nests (with $160\,\text{km}$ resolution) and one simulation with the nests (with 160, 80 and $40\,\text{km}$ resolution) as visualised in Fig. 3 including two-way nesting.

STS particles are considerably smaller than ice and NAT particles (e.g., Considine et al., 2000) and hence $\text{Kn} \gg 1$ so that the slip flow correction term in Eq. (9) can be neglected, which yields:

$$k_{\text{het},h,\text{STS}} = \frac{\gamma_{h,\text{STS}}\overline{v}_{i(h)}S_{\text{STS}}}{4N_{j(h)}} \tag{10}$$

The total reaction rate constant is the sum of the reaction rate constants on all three types of PSCs. The uptake coefficients in ICON-ART are used either from Carslaw et al. (1995) or Sander et al. (2011b). A summary of the $\gamma$ values can be found in Table A2 of Appendix A. They also compare well to other models (e.g., Solomon et al., 2015).

## 3 Simulation with nests around the Antarctic Peninsula

In order to compare the results of the PSC scheme in ICON-ART with satellite measurements and to investigate the impact of the nesting technique on mountain-wave induced PSCs, a three-step simulation is conducted, see Fig. 2. The meteorological variables are initialised with the reanalysis product of the European Centre for Medium-Range Weather Forecasts (ECMWF) ERA-Interim (Dee et al., 2011) on 01 March 2008. The first of March is chosen because almost no PSCs are formed at this time either in the northern or in the southern hemisphere. In 2008, a mountain-wave event took place between 19 and 29 July around the Antarctic Peninsula (Noel and Pitts, 2012) which is further investigated in this study. Sea surface temperature and sea ice cover are based on monthly varying values of the climatology by Taylor et al. (2000), linearly interpolated to the simulation date. The advective model time step is set to $360\,\text{s}$. Vertically, the same 90 levels are used as in the operational setup of DWD weather forecasts, covering the altitude range from the surface up to $75\,\text{km}$ (see e.g., Weimer et al., 2017, Fig. 1). In the lower stratosphere, the vertical grid spacing increases from $400\,\text{m}$ at an altitude of $12\,\text{km}$ up to about $1200\,\text{m}$ at $30\,\text{km}$.

The chemical trace gases (tracers) are initialised by an EMAC simulation which included tropospheric as well as stratospheric chemistry similar to Jöckel et al. (2016). NAT PSCs are simulated with the kinetic NAT parametrisation. The size distribution is shown in Fig. 1. We prescribe $H_2SO_4$ in the lower stratosphere (Thomason et al., 2008; SPARC, 2013).

The emission data sets used in this study are summarised in Table 1. Emissions of $CH_4$, CO, $CO_2$, $N_2O$, $SO_2$ and $CFCl_3$ are considered. The GEIA dataset for chlorofluorocarbons (CFCs) provided by the Emissions of atmospheric Compounds and



**Table 1.** Emission datasets used in this study.

| Species | GEIA[a] | MACCity[b] | MEGAN-MACC[c] | GFED3[d] | EDGARv4.2[e] |
|---------|---------|------------|---------------|----------|--------------|
| $CH_4$ | – | ✓ | ✓ | ✓ | – |
| CO | – | ✓ | ✓ | – | ✓ |
| $CO_2$ | – | – | – | ✓ | ✓ |
| $N_2O$ | – | – | – | ✓ | ✓ |
| $SO_2$ | – | ✓ | – | ✓ | – |
| $CFCl_3$ | ✓ | – | – | – | – |

[a] Cunnold et al. (1994)

[b] van der Werf et al. (2006); Lamarque et al. (2010); Granier et al. (2011); Diehl et al. (2012)

[c] Sindelarova et al. (2014)

[d] van der Werf et al. (2010)

[e] Janssens-Maenhout et al. (2011, 2013)

Compilation of Ancillary Data base (ECCAD) includes only the year 1986 and should eventually be adapted by the online emission tool by Jähn et al. (2020) for the simulated year. This is why other CFCs, apart from $CFCl_3$, are neglected in the less than 1-year simulation (see below) of this study which is therefore dominated by the initialisation of the chemical species. In the context of the recently found source of CFCs (Montzka et al., 2018; Lickley et al., 2020), further CFCs should be considered in future simulations.

In the first step, the simulation starts with a global resolution of R2B04 ($\Delta x$ of about $160\,km$) as a free-running simulation until 01 May 2008. This period is used as a spin-up period for the chemistry until the southern hemispheric polar vortex begins to form. The resolution of $160\,km$ is comparable to current chemistry climate models (e.g., Morgenstern et al., 2010; Kirner et al., 2015; Zhu et al., 2015; Steiner et al., 2020). Thus, this simulation shows how mountain-wave induced PSCs can be treated in a resolution that is similar to other chemistry climate models.

In the second step, it is of importance that the dynamics of the atmosphere and especially the polar vortex are represented realistically in the model, especially for the evaluation with measurement data. This is why the meteorological variables are reinitialised every second day by ERA-Interim in the period between 01 May and 18 July 2008, but the chemical tracers are free-running. This method was already introduced e.g. in Schröter et al. (2018).

In the third step, to avoid the determination of the simulations by lower resolution models and models with hydrostatic

dynamical core, two different free-running simulations are conducted during the mountain wave event from 19 to 29 July 2008: one simulation without any nests and one simulation with nests around the Antarctic Continent (R2B05, $\Delta x$ of about $80\,km$) and around the Antarctic Peninsula (R2B06, $\Delta x$ of about $40\,km$), see Fig. 3 and Table 2. Output in this part of the simulation is given (1) at the specific Antarctic Peninsula overpasses of AIRS for the Antarctic Peninsula nest and (2) hourly during the whole period of the mountain wave event.

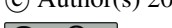



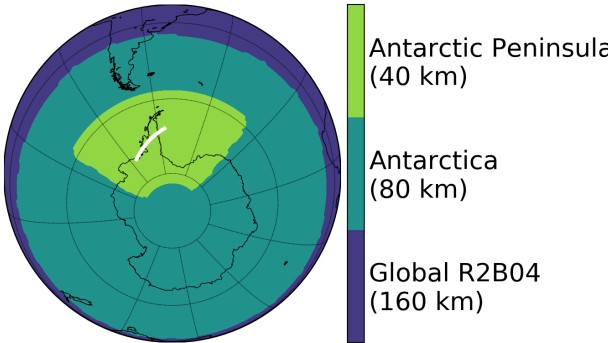

**Figure 3.** Visualisation of the nested domains used in the simulation with the nests: A global resolution of R2B04 ($\Delta x \approx 160\,\mathrm{km}$) is used, with first circular refined grid around the Antarctic Continent (R2B05, $\Delta x \approx 80\,\mathrm{km}$) and a second rectangular refinement around the Antarctic Peninsula (R2B06, $\Delta x \approx 40\,\mathrm{km}$). The white line shows the location of the cross section analysed in Sect. 5.4.

**Table 2.** Simulation setup: The first two months from 01 March to 30 April 2008 are a spin-up period for the chemical tracers. Then, the dynamical variables are reinitialised every second day until 18 July 2008. During the mountain wave event, two simulations are performed: one simulation without any nests and one simulation with the nests. Output in the simulation with nests is hourly and at the specific Antarctic Peninsula overpasses of AIRS analysed in Sect. 5.3.

| Time period | Resolution | Output interval (h) | Remark |
|---|---|---|---|
| 01 March – 30 April | R2B04 | 24 | Spin-up for chemistry, free-running |
| 01 May – 18 July | R2B04 | 24 | Dynamics reinitialised every second day |
| 19 July – 29 July | R2B04 | 1 | Free-running |
| | see Fig. 3 | see main text | Including two-way nesting |

## 4 Satellite datasets

### 4.1 CALIOP

The Cloud-Aerosol LIdar with Orthogonal Polarisation (CALIOP, Pitts et al., 2009; Höpfner et al., 2009; Pitts et al., 2018) onboard the Cloud-Aerosol Lidar and Infrared Pathfinder Satellite Observation (CALIPSO) was launched on 28 April 2006 (Winker et al., 2007). In 2008, the satellite flew as part of NASA's A-train constellation in an orbit with $98°$ inclination at an altitude of $705\,\mathrm{km}$ (Stephens et al., 2002; Pitts et al., 2018). Its orbit was sun-synchronous with equator crossings at about 01:30 and 13:30 LT. It measured down to $82°$S with a repeat cycle of 16 days. In February 2018, it was moved to a lower orbit.

CALIOP is a light detecting and ranging (Lidar) instrument, set up in nadir geometry. It scans the atmosphere at wavelengths of 532 and $1064\,\mathrm{nm}$ with parallel and orthogonal polarisations for the 532-nm channel (Winker et al., 2007). In altitudes from $8.4$ to $30\,\mathrm{km}$, the vertical resolution is $180\,\mathrm{m}$ or higher. At the Earth's surface, the light beam has a diameter of about $100\,\mathrm{m}$ (Höpfner et al., 2009; Pitts et al., 2018).





For the detection of PSCs, a combination of two values is used (Pitts et al., 2018): (1) the ratio of total and molecular backscatter coefficient $R_{532}$ and (2) the backscatter coefficient at perpendicular polarisation $\beta_{532}$, both at a wavelength of $532\,\mathrm{nm}$. Discrimination of the PSC types can be seen in diagrams of $R_{532}$ vs. $\beta_{532}$. Different regions in this diagram refer to the different PSC types (see Pitts et al., 2018).

We use the PSC climatology of Pitts et al. (2018), which is the version 2 level 1B data, averaged with a window of $5\,\mathrm{km}$ along the satellite path. It is restricted to night-time southern hemispheric data during the mountain wave event (data is available within the time frame we are interested in from 22 to 29 July 2008). We examine the altitude levels between 15 and $30\,\mathrm{km}$ that are relevant for formation of PSCs on the one hand and exclude tropospheric clouds on the other hand. In addition, the dataset includes temperature and pressure data of Modern-Era Retrospective analysis for Research and Applications version

2 (MERRA2, Gelaro et al., 2017), which originally has a horizontal resolution of $0.5° \times 0.625°$, interpolated on the CALIOP paths.

## 4.2   AIRS

The Atmospheric InfraRed Sounder (AIRS, Aumann et al., 2003; Chahine et al., 2006) is one of the instruments onboard the Aqua satellite, which was launched in May 2002 and which is part of the A-train constellation right ahead of CALIPSO. Thus,

its orbit is the same as CALIPSO in 2008 but about 1 to $2\,\mathrm{min}$ ahead of CALIPSO[1].

     The AIRS instrument is a nadir sounder with across-track scanning capabilities that scans the atmosphere by 90 footprints per scan with a ground coverage of $1780\,\mathrm{km}$ and a size of $13.5 \times 13.5\,\mathrm{km}^2$ (nadir) to $41 \times 21.4\,\mathrm{km}^2$ (scan edge) per footprint (e.g., Orr et al., 2015; Hoffmann et al., 2017). It measures the spectrally resolved radiances in wavelengths between 3.74 and $15.4\,\mathrm{\mu m}$. Brightness temperatures in the $15\,\mathrm{\mu m}$ band can be used to derive information about gravity waves in the lower

polar stratosphere (Hoffmann et al., 2017). In this study we use a data product averaging over 21 channels around $15\,\mathrm{\mu m}$ to improve the signal-to-noise ratio (Hoffmann et al., 2017). The temperature weighting function in this band peaks at an altitude of around $23\,\mathrm{km}$ with a full width at half maximum of $15\,\mathrm{km}$ and with information from the altitude range between 17 and $32\,\mathrm{km}$. Therefore, it is well suited to derive information about gravity waves in the altitude region where PSCs are expected to exist.

In this study, this band is used to examine the mountain wave event at the Antarctic Peninsula mentioned above. The same algorithm as in previous studies is used to compare the model data with specific Antarctic Peninsula overpasses of AIRS (Hoffmann and Alexander, 2010; Hoffmann et al., 2016, 2017). In particular, the ICON-ART data are resampled on the AIRS measurement grid and a radiative transfer model is used to simulate AIRS measurements based on the ICON-ART data to allow for a direct comparison with the real observations.

---

[1]https://www-calipso.larc.nasa.gov/about/atrain.php, last access 05 November 2020





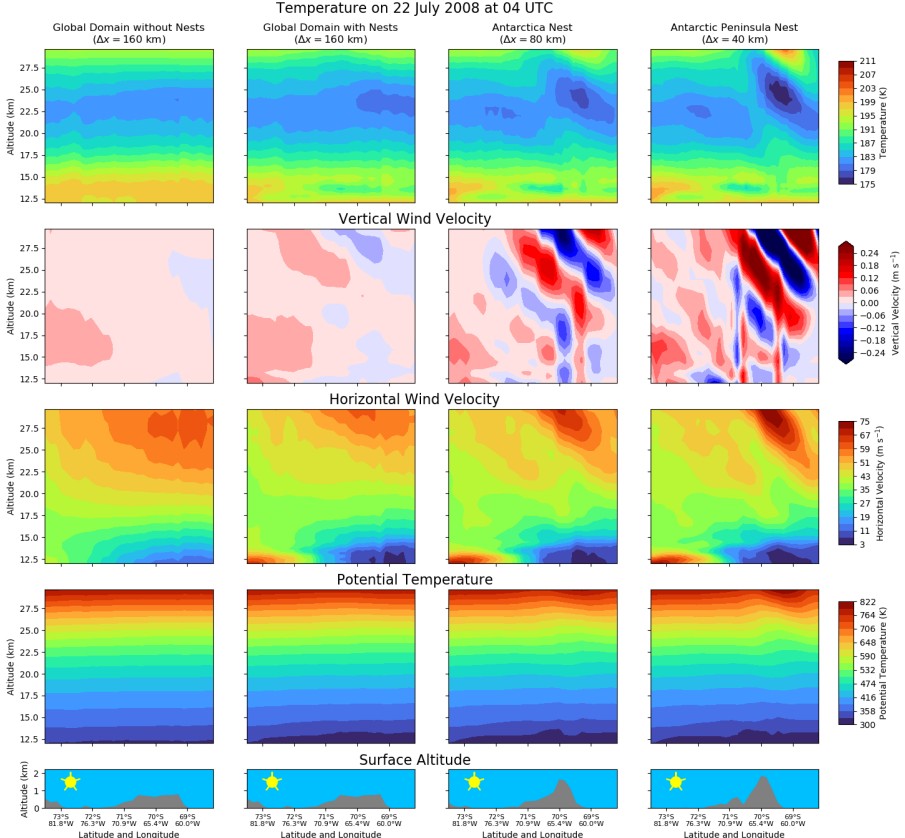

**Figure 4.** Cross sections on 22 July 2008 at 04 UTC along the white line in Fig. 3. Each row represents a different variable in the model. The bottom row shows the surface altitude with the Antarctic Peninsula at around $65°$W. The columns represent the different simulations and domains: The first column is the simulation without nests ($\Delta x \approx 160\,\mathrm{km}$), the second one is the global domain ($\Delta x \approx 160\,\mathrm{km}$) including two-way nesting and the two right columns represent the Antarctica ($\Delta x \approx 80\,\mathrm{km}$) and Antarctic Peninsula nests ($\Delta x \approx 40\,\mathrm{km}$), respectively. The dynamical variables temperature, vertical wind velocity, horizontal wind velocity and potential temperature are shown in this figure.

## 5 Mountain-wave induced PSCs with ICON-ART


In this study, we investigate a mountain wave event during July 2008 with ICON-ART in a configuration with interactive chemistry and local grid refinement around the Antarctic Peninsula. This section comprises an evaluation of the dynamical structure (Sect. 5.1), comparisons of the model results with CALIOP and AIRS measurements (Sect. 5.2 and Sect. 5.3) and discusses the impact of the direct simulation of mountain-wave induced PSCs on the polar ozone chemistry (Sect. 5.4).



## 5.1 Dynamical structure of the mountain wave on 22 July 2008

Figure 4 shows cross sections of the mountain wave dynamics along the line shown in Fig. 3 for the example of 22 July 2008 at 04 UTC in the altitude range between 12 and 30 km. The model data is interpolated to the path by an inverse distance method including the three neighboured grid points.

This figure demonstrates that the mountain wave can be directly simulated with the resolution of 40 km. The left column shows the simulation without nests whereas the three other columns illustrate the dynamics in the different domains shown in Fig. 3 of the simulation with the nests.

The bottom row of Fig. 4 shows the resolution of the Antarctic Peninsula which is a crucial parameter for simulating mountain-wave induced PSCs as it determines the flow over the mountain range. As can be seen, in the global domains with resolution of 160 km the top of the Antarctic Peninsula does not exceed 1 km. Therefore, the flow over the mountain range is underrepresented in the model. By increasing the resolution, however, to 40 km the height of the Antarctic Peninsula peaks at around 2 km and can be represented as mountain range with associated directly simulated flow over the orography.

This is reflected in the variables shown in Fig. 4. Temperatures as low as 175 K only occur in the Antarctic Peninsula nest with a 40 km resolution in the lee of the mountain. In addition, it shows the characteristic high and low temperature patterns as calculated by theory (e.g., Queney, 1947) and seen by measurements (e.g., Wright et al., 2017). The temperature perturbation in the order of 10 K is consistent with previous studies of mountain waves (Meilinger et al., 1995; Carslaw et al., 1998; Eckermann et al., 2009).

The horizontal wavelength of the simulated mountain wave is in the order of 240 km, which is a medium-large wavelength compared to other events (see e.g., Alexander and Teitelbaum, 2007; Plougonven et al., 2008; Hoffmann et al., 2014). In addition, the vertical resolution is in the order of 500 m at least in the lower parts of Fig. 4. This is why the mountain wave can be captured by the resolution of 40 km in the Antarctic Peninsula nest.

The relatively large temperature gradients in the Antarctic Peninsula nest are also in agreement with gradients in the wind velocities which are shown in the second and third row of Fig. 4. In the lee of the Antarctic Peninsula, the vertical wind velocity changes signs in altitudes where the temperature increases or decreases and has maximum values above $0.3\,\mathrm{m\,s^{-1}}$. Largest horizontal wind velocities in the order of $75\,\mathrm{m\,s^{-1}}$ occur in altitudes above 27 km.

The mountains also cause perturbations in the potential temperature (forth row in Fig. 4) in the lee. If diabatic processes are negligible, the flow follows the contours of potential temperature which will be shown in the PSC precursors in Sect. 5.4.2.

Up to now, it was only demonstrated that mountain waves are directly simulated in the Antarctic Peninsula nest (right column) and are consistent with the theory of mountain waves. In addition, Figure 4 also shows the impact of the two-way nesting in ICON-ART. The mountain wave at the Antarctic Peninsula cannot be represented adequately in the resolution of 160 km in the simulation without the nests (left column in Fig. 4). The characteristic wave patterns do not occur in this simulation. In contrast to this, the global domain in the simulation with the nests (second column in Fig. 4) shows a decrease in temperature of about 2 K in the lee of the mountains as a result of the two-way nesting and the lower temperature due to





the directly simulated mountain wave in the Antarctic Peninsula nest. Of course, the amplitude is lower than it is expected by mountain waves but the effect of the mountain wave is still remarkable also in the global grid of 160 km resolution.

This is also visible in the other variables of Fig. 4. Especially for the vertical wind velocity (second row), one can see that wave-like structures occur in the global domain where they cannot be represented without the nests. Therefore, we can expect that mountain-wave induced PSCs can also be represented in this relatively low global resolution because they are directly simulated in the locally refined regions.

## 5.2 Comparison of simulated PSCs with CALIOP measurements

For a comparison with CALIOP, the ICON-ART PSC volume concentrations of the Antarctic Peninsula nest are interpolated (1) horizontally by an inverse distance method, (2) linearly in time and (3) linearly in geometric altitude to all CALIOP paths from 22 to 29 July 2008 where CALIOP's orbit was within the region of Antarctic Peninsula nest. Details about the interpolation can be found in Weimer (2019, Sect. 4.7.4).

As pointed out by previous studies, an adequate comparison of CALIOP with model data can only be set up if the model 300 data is transferred into the optical space measured by CALIOP at 532 nm using the method by Engel et al. (2013),Tritscher et al. (2019) and Steiner et al. (2020), which we apply here. It is based on T-matrix and Mie calculations (e.g., Mishchenko et al., 1996) with particle number densities and particle radii of the PSC types as input and parallel as well as perpendicular backscatter coefficients and backscatter ratio as output. In accordance with Tritscher et al. (2019) and Steiner et al. (2020) we use aspect ratios of ice and NAT of 0.9 and refractive indices of 1.31 for ice and 1.48 for NAT.

As described in Pitts et al. (2018), the thresholds to determine the boundaries between STS, NAT and ice are dynamically calculated depending on the measured values. Like in Steiner et al. (2020), we calculate the thresholds in the backscatter ratio ($R_{532,\text{thres}}$) and perpendicular backscatter coefficient ($\beta_{\perp,\text{thres}}$) as daily means from data in the Antarctic Peninsula nest. We add the measurement uncertainty to these thresholds to determine the PSC types as statistical outliers as in the measurements based on Engel et al. (2013) and Tritscher et al. (2019). Finally, the backscatter ratio threshold between NAT and ice ($R_{\text{NAT,ice}}$) 310 depends on the concentrations of gaseous $H_2O$ and $HNO_3$, thus on the state of denitrification and dehydration. We use the average threshold of 2.75 suggested by Pitts et al. (2018) because the data structure is similar in space as well as in time to the data used for Fig. 5b of Pitts et al. (2018).

Since the nested simulation is free-running we cannot expect the PSCs to occur at exactly the same locations as in the measurements. Therefore, we compare the development of PSCs with respect to the temperature in both datasets, which is a 315 crucial parameter in the formation of PSCs (e.g., Solomon, 1999). We count the occurrence of the different PSC types by the method described above in temperature bins and compare them between ICON-ART and CALIOP.

The results can be found in Fig. 5. It shows the relative number of grid points (in %) on the CALIOP paths where the different PSC types occur. The "No" PSC category is obviously the dominant category for both ICON-ART and CALIOP, but neglected in this comparison to see only the occurrence of PSCs in the temperature bins.

In Figure 5, the size of the circles correspond to the total number of grid points with PSCs, that is also printed on the right hand side of the panels, and the colour of the circles show the relative number of this PSC type occurrence in the respective





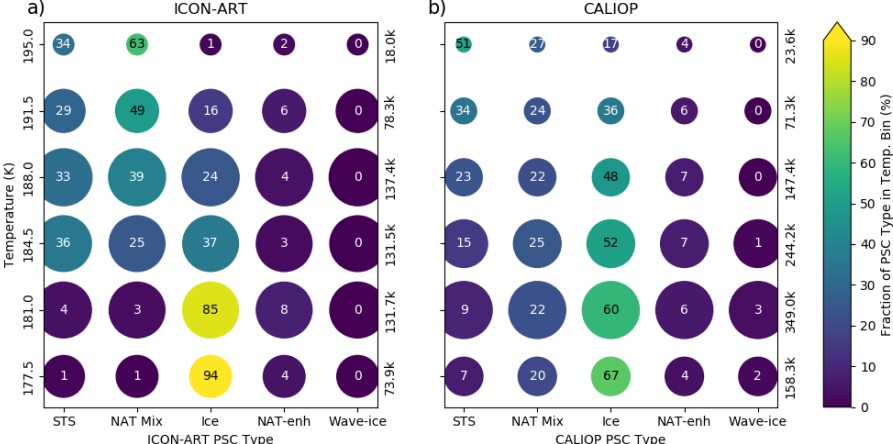

**Figure 5.** Statistical analysis of PSC occurrence for a) ICON-ART simulation and b) CALIOP measurements in the Antarctic Peninsula nest. The ICON-ART data is interpolated to the CALIOP paths in the altitude range from 15 to 30 km in the nest in the time range between 22 and 29 July 2008 where CALIOP data is available. The colours correspond to the numbers in the circles and show the fraction of the PSC type that is present in the temperature bins with a width of 3.5 K. The sizes of the circles correspond to the total number of grid points with PSCs in the temperature bins relative to the maximum in each panel, also denoted at the right hand side of the panels. The temperature data for CALIOP originates from MERRA2, which is part of the CALIOP product.

temperature bin. For ICON-ART (panel a), the modelled temperatures are used directly whereas the temperature for CALIOP (panel b) is interpolated from MERRA2 and provided as part of the original dataset (Pitts et al., 2018).

As can be seen, ice particles dominate the existence of PSCs in the lowest temperature bins of 181 and 177.5 K for both

ICON-ART and CALIOP. In addition, the "NAT-enh" category is negligible for both datasets. The development of pure STS particles is similar between ICON-ART and CALIOP, although the fractions are underestimated at high temperatures. However, the number of measurements in this bin is small compared to the other ones so that this statistic could be non-significant. Whereas NAT particles in the "NAT Mix" category exist over the whole temperature range with a similar fraction in the measurements, their existence is cut off at temperatures lower than about 183 K, the lower boundary of the 184.5 K bin. This

is due to the used operator splitting and will be further investigated in Sect. 5.4.

In addition to the analysis of direct temperature as reference, we also analyse the development of PSCs in the same manner relative to $T_{\mathrm{NAT}}$ (Fig. 6) and $T_{\mathrm{ice}}$ (Fig. 7), calculated with $X_{\mathrm{H_2O}} = 5\,\mathrm{ppmv}$ and $X_{\mathrm{HNO_3}} = 10\,\mathrm{ppbv}$ as input. We chose these constant values because the mixing ratios measured by Microwave Limb Sounder (MLS), also provided as part of the CALIOP product, only include the gaseous part of $\mathrm{HNO_3}$ and $\mathrm{H_2O}$ which leads to inconsistencies when knowing that PSCs exist in

CALIOP. Nevertheless, this analysis includes the pressure dependence of the PSC existence temperatures so that it provides more realistic thresholds for the existence temperatures than constant values like 195 and 188 K.

Although the number of grid points is relatively low for temperatures around $T_{\mathrm{NAT}}$ and higher in Fig. 6, no ice PSCs exist in both simulation (panel a) and measurements (panel b). The major fraction in this temperature region is accounted to the "NAT

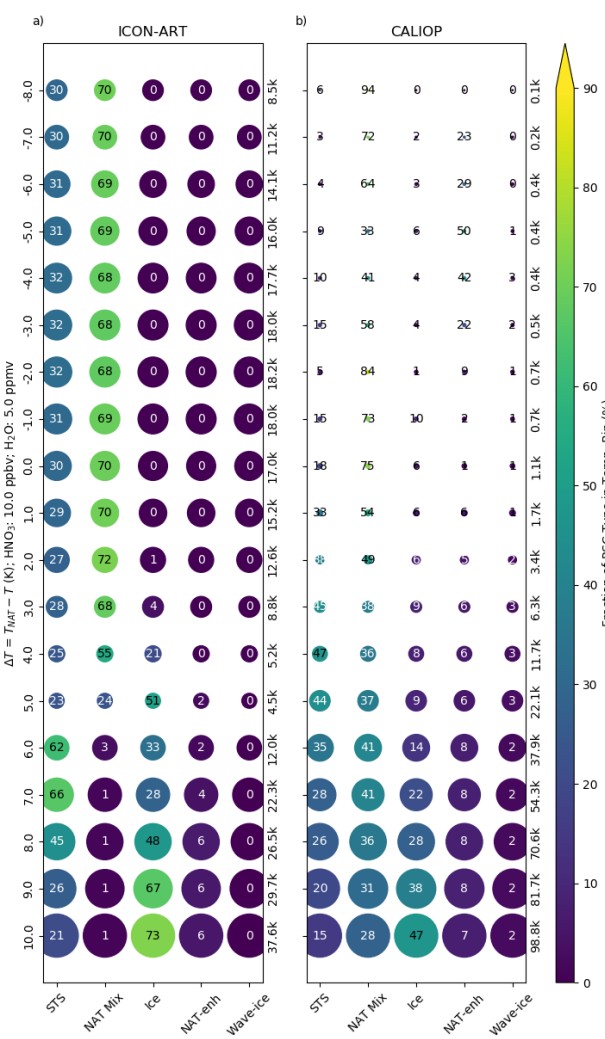

**Figure 6.** Statistical analysis of PSC occurrence for a) ICON-ART simulation and b) CALIOP measurements relative to $T_{\mathrm{NAT}}$. Here, a common $T_{\mathrm{NAT}}$ for both y-axes is derived from Hanson and Mauersberger (1988) with input of $X_{\mathrm{H_2O}} = 5\,\mathrm{ppmv}$ and $X_{\mathrm{HNO_3}} = 10\,\mathrm{ppbv}$. The data are binned in terms of temperature difference with a bin width of $1\,\mathrm{K}$. Apart from that, the structure is the same as in Fig. 5.



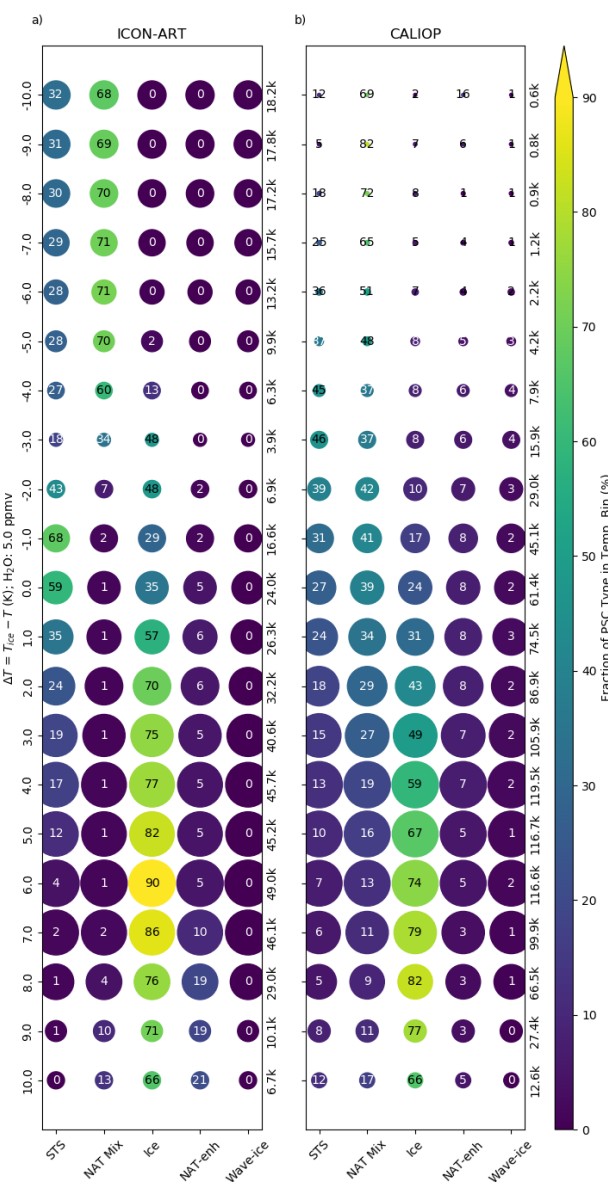

**Figure 7.** Statistical analysis of PSC occurrence for a) ICON-ART simulation and b) CALIOP measurements relative to $T_{ice}$. Here, a common $T_{ice}$ for both y-axes is derived from Marti and Mauersberger (1993) with input of $X_{H_2O} = 5\,\mathrm{ppmv}$. The data are binned in terms of temperature difference with a bin width of $1\,\mathrm{K}$. Apart from that, the structure is the same as in Fig. 5.





mix" category in both panels although PSCs usually do not exist. In total, the number of data points in ICON-ART is higher

for $T > T_{\mathrm{NAT}}$ and lower for $T < T_{\mathrm{NAT}}$ in comparison to CALIOP. However, the principal development is similar to CALIOP: at temperatures about $5\,\mathrm{K}$ lower than $T_{\mathrm{NAT}}$ ice PSCs begin to form and turn out to be the major fraction of all PSC types. As already mentioned in the description of Fig. 5, NAT PSCs are cut off at a certain point, which will be investigated later in this study.

The upper part of Fig 7, which shows the PSC development relative to $T_{\mathrm{ice}}$, corresponds to the lower part of Fig. 6 with a

shift of about $8\,\mathrm{K}$. In accordance to the measurements, the fraction of ice PSCs in ICON-ART start to dominate at temperatures lower than $T_{\mathrm{ice}}$ with fractions larger than $50\,\%$. The largest fractions occur at temperatures about 6 to $7\,\mathrm{K}$ lower than $T_{\mathrm{ice}}$ in both ICON-ART and CALIOP.

Although we will show in the next sections that mountain-wave induced PSCs can be directly simulated in the Antarctic Peninsula nest, no "Wave-ice" PSCs are simulated by ICON-ART. As shown in Eq. (1), the ice particle number concentration

is essentially set to the constant value of $0.25\,\mathrm{cm}^{-3}$. The ice number concentration in mountain waves can increase to values of a few $\mathrm{cm}^{-3}$ and then leading to larger backscatter ratios (e.g., Engel et al., 2014). Therefore, the backscatter ratio to determine the PSC types does not get as large as 50 which is needed for the "Wave-ice" category.

In total, we showed in this section that, apart from some differences in NAT at low temperatures and the "Wave-ice" category, the general formation of PSCs with respect to different reference temperatures is similar to the CALIOP measurements.

Therefore, we will show the development of mountain-wave induced PSCs in the next sections qualitatively but a quantitative comparison is difficult because of the too large particle number concentration of ice PSCs.

### 5.3    Direct simulation of mountain waves compared to AIRS

For the comparison with AIRS, temperature and pressure of ICON-ART in the Antarctic Peninsula nest is saved at the time step closest to each of the AIRS overpasses during 20 and 21 July 2008. These data are then convolved with the same temperature

weighting functions that apply for the AIRS observations (see e.g., Hoffmann et al., 2017). The resulting brightness temperature (BT) perturbations can be found in Fig. 8 on 20 July and in Fig. 9 on 21 July 2008 for AIRS and ICON-ART.

Horizontal structures of the BT perturbations are shown in the first and second columns for AIRS and ICON-ART, respectively. Largest perturbations are present directly above the Antarctic Peninsula for both AIRS and ICON-ART which demonstrates that the perturbations originate from mountain waves propagating into the lower stratosphere. In addition, the

mountain wave has an angle with respect to the Antarctic Peninsula mountains of about $45\,^{\circ}$ suggesting a horizontal wind from south west. This is also represented in the ICON-ART simulation.

Fine structures as e.g. in panel j of Fig. 8 cannot be simulated in this simulation setup of ICON-ART (e.g. panel k) since the resolution of $40\,\mathrm{km}$ is still too coarse to predict them. Thus, some fine structures are missing in the ICON-ART simulation but the general behaviour of the mountain wave can be directly simulated in the Antarctic Peninsula nest.

This is also stressed by the comparison of the perturbations at the latitude of $70\,^{\circ}\mathrm{S}$ that are shown in the third column of Figs. 8 and 9. The largest BT perturbations can be found at the longitude of the Antarctic Peninsula in all the panels. Some fine structures are missing in ICON-ART. For instance, at some overpasses the amplitude of the wave is underestimated (e.g. panel

**Figure 8.** Comparison of AIRS and ICON-ART brightness temperature (BT) perturbations at wavelengths of $15\,\mu m$ for all Antarctic Peninsula overpasses of AIRS during 20 July 2008. The first column shows the BT perturbation observed by AIRS, the second column the simulated perturbation based on ICON-ART in the Antarctic Peninsula nest. The third column shows the perturbation for AIRS (black) and ICON-ART (orange) at a latitude of $70\,°S$. The rows show different overpasses at (a-c) 03:49 UTC, (d-f) 05:28 UTC, (g-i) 18:55 UTC and (j-l) 20:33 UTC.





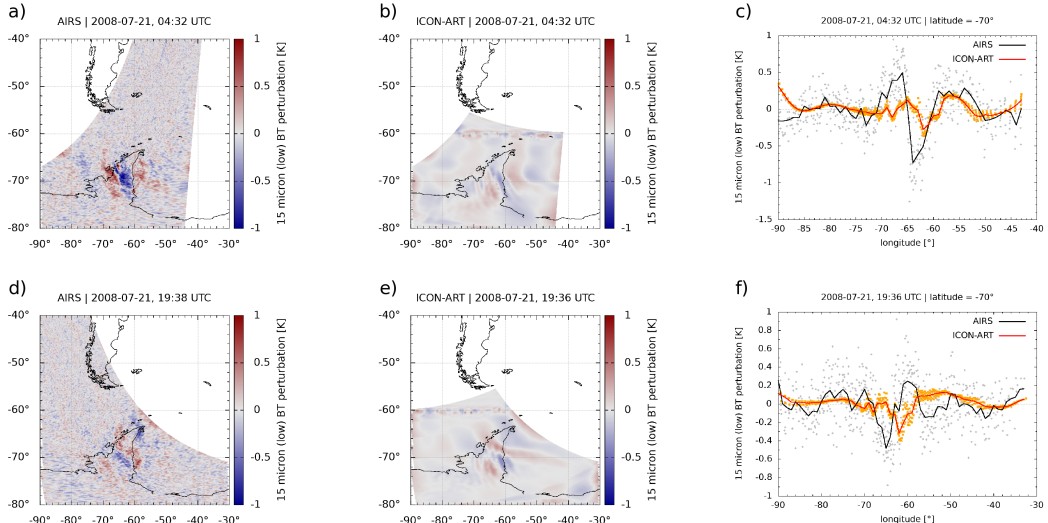

**Figure 9.** Same as Fig. 8 but for 21 July 2008. The rows show different overpasses at (a-c) 04:32 UTC and (d-f) 19:38 UTC.

c of Fig. 9). At other overpasses, the phase of the wave is shifted with respect to AIRS, as e.g. in panels c of Fig. 8 and f in Fig. 9. This is most probably a result of the free-running simulation where the wave cannot be expected to be located at exactly the same location as in the measurements.

In total, we have shown in this section that, apart from some missing fine structures, the mountain wave event taking place in the end of July 2008 can be represented with the resolution of $40\,\mathrm{km}$ in comparison with AIRS.

## 5.4 Impact of mountain-wave induced PSCs on the chemistry

In the previous sections, it was demonstrated that both the PSC formation and the formation of the mountain wave are in relatively good agreement with measurements considering the limits in measurements and simulation setup. In this section, we investigate the impact of directly simulated mountain-wave induced PSCs on the interactively calculated chemistry in ICON-ART.

### 5.4.1 The formation of PSCs in the mountain wave

Figure 10 demonstrates that the gap between direct simulations of mountain-wave induced PSCs and their treatment in relatively coarse global resolutions can be closed by ICON-ART. The STS volume concentration (first row) in the global domain with nests is influenced by the mountain wave, especially at altitudes higher than $20\,\mathrm{km}$ where particle volume concentrations close to zero occur in the global domain which do not exist without the nesting technique. The influence of the mountain





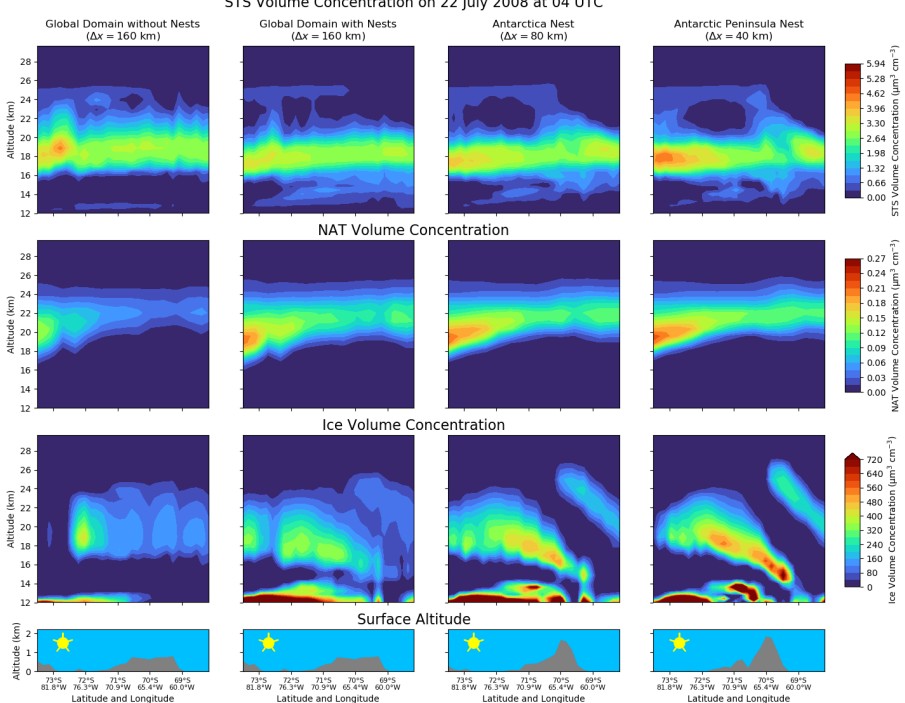

**Figure 10.** Same as Fig. 4 but for the volume concentrations of the different PSCs: STS, NAT and ice. Please note the different colour bars.

wave on STS is amplified within the nests. The STS particles are assumed to freeze at temperatures $3\,\mathrm{K}$ below the frost point (Carslaw et al., 1995; Koop et al., 2000). Thus, STS particles are only computed for higher temperatures so that STS particles

are formed in the mountain wave where the temperature is higher than this threshold.

STS and NAT compete with each other in taking up $HNO_3$. This is why in this case study distinct layers exist: NAT PSCs at altitudes higher than $20\,\mathrm{km}$ and STS PSCs at lower altitudes where also $H_2SO_4$ is enhanced. The NAT volume concentrations are increased in the Antarctic Peninsula nest (second row, right column) which is why they are also increased in the global domain in the simulation with the nests.

In contrast to the literature, the NAT volume concentration decreases when the air masses approach the mountain wave. Since the NAT size bins are advected with the general air masses, the wave-like patterns occur in both nests. As a result of the operator splitting used in ICON-ART, the largest fraction of gaseous $H_2O$ leads to ice formation at temperatures lower than about $180\,\mathrm{K}$ and is not available for NAT and STS PSCs in the mountain wave anymore. Therefore, the largest signal of mountain-wave induced PSCs can be found in ice PSCs in Fig. 10. This is an issue of further investigation in the future. In

addition, NAT PSCs are formed by freezing STS particles in the mountain wave, as shown e.g. by Bertram et al. (2000) and Salcedo et al. (2001). This is not integrated in the model so far and should be considered in the future.

The best example of the formation of mountain-wave induced PSCs are ice PSCs (third row in Fig. 10). In the lee of the Antarctic Peninsula ice PSCs occur with volume concentrations as large as $700\,\mathrm{\mu m^3 cm^{-3}}$. These relatively high values





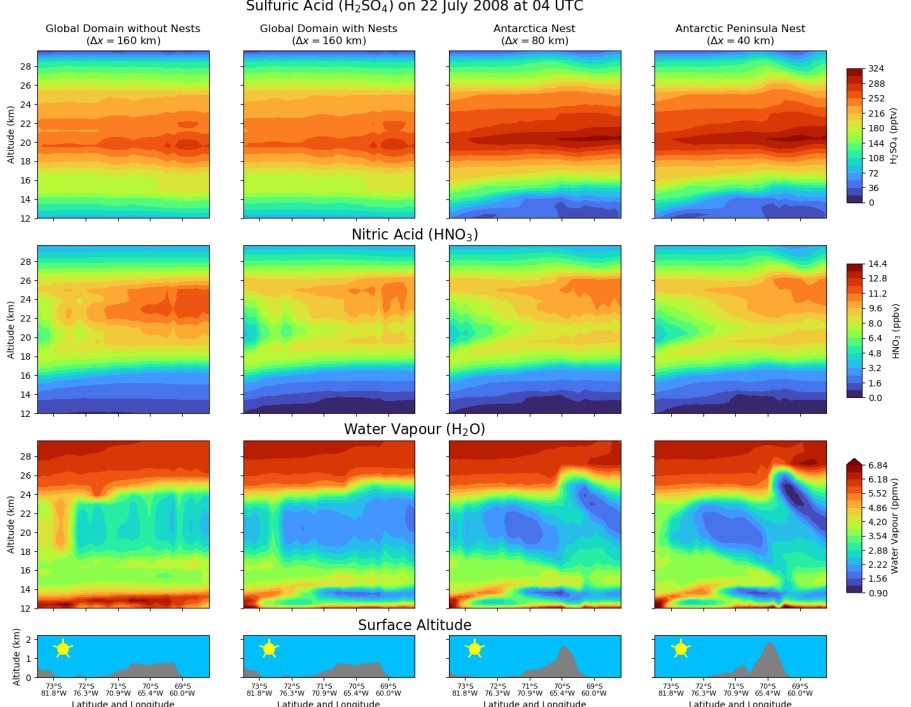

**Figure 11.** Same as Fig. 4 but for the tracers that are relevant for the formation of PSCs: $H_2SO_4$, $HNO_3$ and $H_2O$. Please note the different colour bars.

might be overestimated since the ice number concentration is set to the tropospheric value of $0.25\,\mathrm{cm}^{-3}$ which is too large in

comparison to measurements, as discussed previously (see also Buchholz, 2005, for an overview).

On the other hand, the ice PSCs are clearly connected to the regions where temperature is decreased and show similar wave-like patterns as the temperature. These increased volume concentrations are also present in the global domain where wave-like patterns can be simulated with the nests in contrast to the simulation without the nests where these structures do not exist.

Therefore, this figure shows that mountain-wave induced PSCs can be directly simulated with ICON-ART and their effect

can also be treated in the global domain where mountain-wave induced PSCs cannot be represented without the nests.

### 5.4.2 Influence of the mountain wave on PSC precursors

As mentioned above, long-lived tracers closely follow the potential temperature. Therefore, the wave perturbations in potential temperature can be seen in all tracers shown in Fig. 11 in the lee of the mountain for the simulation with the nests. The wave-like structures in sulfuric acid ($H_2SO_4$, first row), nitric acid ($HNO_3$, second row) and water vapour ($H_2O$, third row) correspond

to the structures seen in the potential temperature (see Fig. 4).





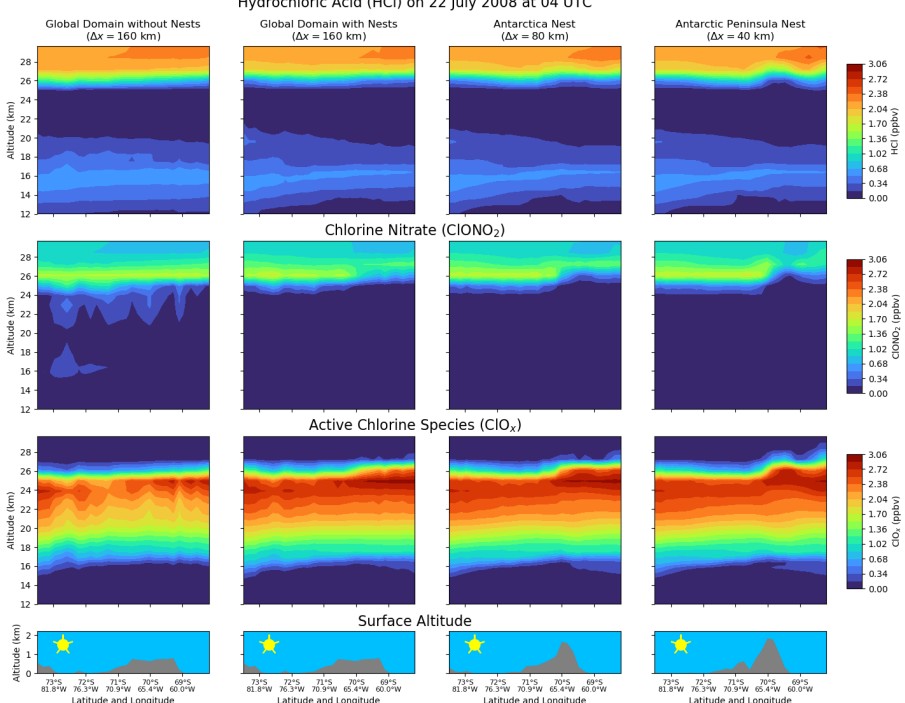

**Figure 12.** Same as Fig. 4 but for the chlorine species: HCl, ClONO$_2$ and the sum of all other (active) chlorine-containing species (ClO$_x$). The colour bars are equal in this figure.

Sulfuric acid (H$_2$SO$_4$) which is prescribed as a climatology in the global domain, but free-running in the nests, only shows a minor signal of the two-way nesting. Due to the missing sink of H$_2$SO$_4$ by sedimentation of aerosols (see Sect. 3), the mixing ratio accumulates in the nested domains (two right columns).

Nitric acid (HNO$_3$), shown in the second row of Fig. 11, is taken up by STS and NAT PSCs so that it is lower in the simulation with nests than in the simulation without the nests. This is shown not only in the nested domains, but can also be returned to the global domain as a result of the two-way nesting.

This property especially occurs for water vapour (H$_2$O) where in the lee at altitudes between 22 and 25 km with temperatures lower than about 179 K volume mixing ratios lower than 1 ppmv occur that are connected with uptake in ice PSCs (cf. Fig. 10). It cannot be represented in the simulation without the nests (left column) where the H$_2$O volume mixing ratio does not decrease to values lower than 2.5 ppmv. In the global domain of the simulation with the nests, however, values as low as 2 ppmv occur in the lee of the mountain as a result of the two-way nesting.

### 5.4.3 Impact of mountain-wave induced polar stratospheric clouds on chlorine activation

The directly simulated mountain-wave induced PSCs in the simulation with the nests also affect the chemistry due to increased heterogeneous reactions on their surface. The chlorine species are summarised in Fig. 12 in the same way as in the previous





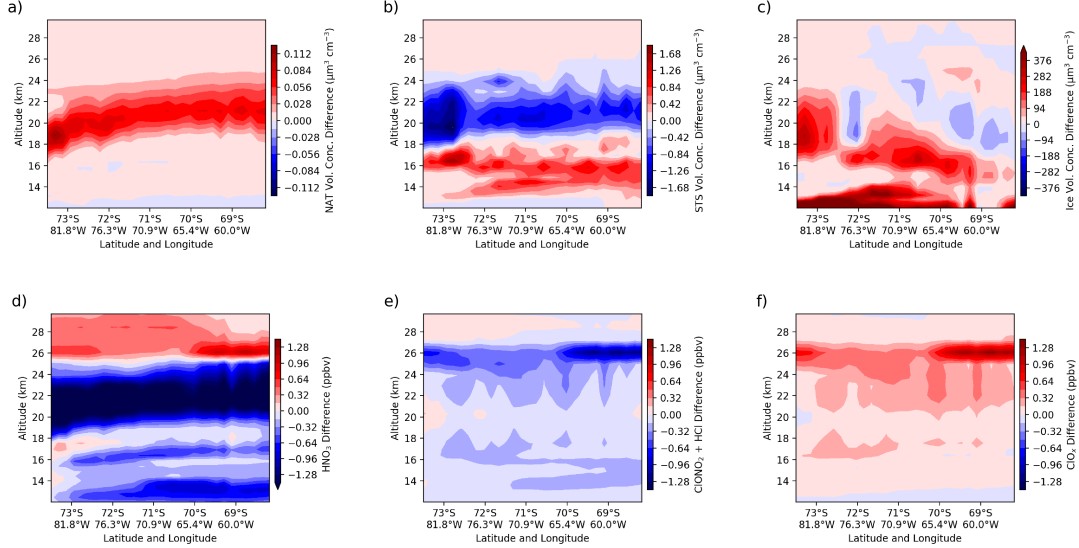

**Figure 13.** Difference in global domain between with and without nest around the Antarctic Peninsula along the cross section shown in Fig. 3 at the same date (22 July 2008, 04 UTC) and using the same algorithm as in the previous figures. The shown variables are differences in (a-c) the NAT, STS and ice volume concentrations, respectively, (d) the $HNO_3$ volume mixing ratio, (e) the volume mixing ratio of the reservoir species $ClONO_2$ and HCl and (f) the $ClO_x$ as defined in Eq. (11). Minimum values of $HNO_3$ are around $2.3\,ppbv$.

figures. The reservoir species HCl (first row) and $ClONO_2$ (second row) are shown together with the active chlorine species (third row), summarised as $ClO_x$:

$$X_{ClO_x} = X_{ClNO_2} + 2\,X_{Cl_2O_2} + X_{OClO} + 2\,X_{Cl_2} + X_{BrCl} + X_{HOCl} + X_{Cl} + X_{ClO} \tag{11}$$

During July, which is a relatively late stage of the southern polar winter, most of the chlorine species in the altitude range between 15 and $25\,km$ have been already activated. The broad band of volume mixing ratios of $ClO_x$ with values up to about

$2.2\,ppbv$ that are present in all panels of the third row suggests this. Therefore, additional chlorine activation can only be expected in altitude regions above $25\,km$. In the previous sections, it was shown that the directly simulated mountain-wave induced ice PSCs reaches altitudes of about $26\,km$ (see Fig. 10). Therefore, we focus on these altitudes in the following analyses.

The mountain-wave induced PSCs lead to increased chlorine activation in the lee of the Antarctic Peninsula that is not

present in the simulation without the nests. At altitudes around $26\,km$, the additional ice PSCs activate both $ClONO_2$ and HCl in the lee of the mountains. This is shown by values of the reservoir species around zero and values of $ClO_x$ up to $3\,ppbv$ in this region that cannot be simulated without the nests.

A summary of the impact of the two-way nesting on mountain-wave induced PSCs and the chemistry can be found in Fig. 13 where differences of the two global domains between with and without the nests are shown for various variables of the





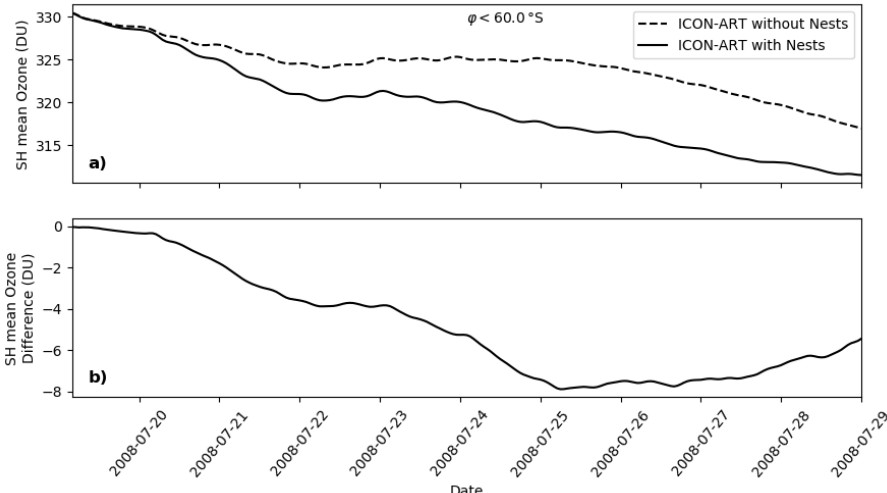

**Figure 14.** Time series of the mean total ozone column during the third period of the simulation in the global domains for latitudes south of 60 °S (panel a). The dashed line is the simulation without the nests and the solid line shows the ozone development in the simulation with the nests. The difference between these lines is illustrated in panel b.

previous sections. Panel a shows the difference in the NAT volume concentration and it can be seen that more NAT particles are produced in the simulation with the nests as a result of the two-way nesting. These enhanced NAT volume concentrations lead to (1) a decreased volume concentration in STS (panel b) as a result of the used operator splitting and (2) to decreased values of $HNO_3$ (panel d) with differences lower than $-1.4\,\mathrm{ppbv}$ in this region since $HNO_3$ is taken up by NAT particles. In regions where no additional NAT particles exist, the STS volume concentration is enhanced due to the two-way nesting (panel 450 b). In the ice volume concentration in panel c, the wave-like structure is clearly visible.

The mountain-wave induced (ice) PSCs lead to additional chlorine activation, in this example in an altitude of about $26\,\mathrm{km}$ where the negative difference in the sum of $ClONO_2$ and $HCl$ (panel e) is closely connected to the positive differences in $ClO_x$ in the lee of the mountain (panel f). $HNO_3$ is also a product of heterogeneous reactions on the surface of PSCs and therefore is enhanced in this region with respect to the simulation without the nests (panel d).

Altogether, it was shown so far that with the local grid refinement around the Antarctic Peninsula mountain-wave induced PSCs can be represented in the global domain where they cannot be simulated without the nests. These PSCs lead to increased chlorine activation in the model.

### 5.4.4 Impact of mountain-wave induced polar stratospheric clouds on ozone

Due the mountain wave event in July 2008 above the Antarctic Peninsula, chlorine activation is increased in the simulation with 460 the nests compared to the simulation without the nests. The impact on ozone cannot be expected to be present at the Antarctic Peninsula itself since the activated chlorine species are transported during the polar night within the polar vortex. Therefore,





ozone is affected by the mountain wave event (1) far in the lee of the mountain and (2) later in the year. This is why the time
series of the southern hemispheric mean total ozone column for the whole third period of the simulation is shown in Fig. 14.
The ozone columns are averaged for latitudes south of $60\,°S$. Two lines are shown in panel a: the solid line is the mean total
ozone column in the global domain of the simulation with the nests, whereas the dashed line illustrates the simulation without
the nests.

As can be seen, differences in the simulations are below $1\,DU$ during the whole day of 19 July (see also panel b). This first
day is a spin-up period for the nests that are initialised with the global domain to form the mountain wave. From 20 July until
end of the simulation, however, the mean ozone column in the simulation with the nests is smaller than in the simulation without
the nests. While ozone is generally decreasing during this period in both simulations, the absolute value of the difference peaks
at around $8\,DU$ on 25 July and decreases afterwards. The higher resolution around the Antarctica and the Antarctic Peninsula
seems to lead to generally lower ozone columns. This additionally highlights the need of higher resolutions in atmospheric
chemistry models.

Figure 14 demonstrates that the higher load of active chlorine species due to the directly and interactively formed mountain-
wave induced PSCs lead to a by up to $8\,DU$ larger decrease of ozone in the southern hemisphere. The prediction skill after
$10\,days$ is fairly low so that an analysis after the shown period is not possible with a free-running simulation. Future investiga-
tions could extend this period with specified dynamics in both simulations to be able to examine the impact of mountain-wave
induced PSCs on the ozone hole during September and October.

## 6 Conclusions and outlook

Seamless modelling of chemistry-climate interactions is challenging. Not many modelling systems can do this in a consistent
way. In the past, it was impossible to directly simulate mountain-wave induced polar stratospheric clouds (PSCs) in global
chemistry models due to the coarse resolution in that kind of simulations. In this study, we investigated this problem with the
scheme for PSCs in the ICOsahedral Non-hydrostatic modelling framework with its extension for Aerosols and Reactive Trace
gases (ICON-ART). The scheme forms ice PSCs based on the microphysics of the meteorological model, STS particles by the
analytic expression of Carslaw et al. (1995) with some improvements with respect to the constant particle number concentration
and NAT particles by a kinetic non-equilibrium approach with a flexibly selectable size distribution.

We performed a three-step simulation to investigate the impact of mountain-wave induced PSCs in ICON-ART on the chem-
istry: First, a free-running simulation was conducted from 01 March to 30 April 2008. Second, the dynamics were reinitialised
every second day by ERA-Interim data to ensure a realistic development of the polar vortex until 18 July 2008. Third, two
free-running simulations followed: a simulation including two-way nesting with nests around the Antarctica ($\Delta x \approx 80\,km$)
and the Antarctic Peninsula ($\Delta x \approx 40\,km$) and a simulation without these nests.

The results were compared with measurements by CALIOP and AIRS in the Antarctic Peninsula nest. The PSC types of
ICON-ART were derived from an algorithm that transfers the PSC volume concentrations to the spectral space of CALIOP
so that a statistical comparison between both datasets could be established. The CALIOP PSC type "Wave-ice" could not be





simulated with the model most probably as a result of the constant ice number concentration set to $0.25\,\mathrm{cm}^{-3}$. The analyses also showed the need of an interactive calculation of PSCs to treat the different PSC types competing with the available gaseous $HNO_3$ and $H_2O$. The comparison with all CALIOP measurements within the Antarctic Peninsula nest demonstrated that the general formation of most of the PSC types in ICON-ART is similar with respect to temperature. At low temperatures, ice PSCs dominate in both datasets whereas STS and NAT types have the largest fractions for temperatures higher than the ice

formation temperature. This was also pronounced by evaluations with respect to the NAT and ice formation temperatures.

The comparison to AIRS demonstrated for all Antarctic Peninsula overpasses of 20 and 21 July 2008 that the main features of the mountain wave can be represented in the resolution of $40\,\mathrm{km}$. The angle of the mountain wave to the mountain range can be represented by the model and the brightness temperature perturbation is in the correct order of magnitude for all overpasses. The investigated mountain wave had a horizontal wavelength of about $240\,\mathrm{km}$ and is therefore a medium large mountain wave

that could be captured by the chosen resolution. For mountain waves with smaller wavelengths, an even higher resolution would be needed to resolve the wave adequately. In addition, even though we had chosen a high resolution, compared to standard climate simulations, we still observed structures that could not be resolved. This underlines the need for high-resolution climate simulations for a suitable physical representation of mountain waves.

By introducing the two-way nesting around the Antarctic Peninsula with resolutions down to $40\,\mathrm{km}$, we were able to directly

simulate the main features of the mountain wave and transfer its effect back to the global domain for a typical event on 19 to 29 July 2008. Thus, additional mountain-wave induced PSCs, formed also in the resolution of $160\,\mathrm{km}$, lead to enhanced chlorine activation and with that to an up to $8\,\mathrm{DU}$ larger ozone depletion above the Antarctic Continent.

Thus, this demonstrates that (1) dynamics, tracers, PSCs and chemistry are interactively and consistently integrated in ICON-ART and that (2) the gap between direct simulations of mountain-wave induced PSCs and their treatment in coarse global

resolutions can be bridged by ICON-ART.

Future simulations will exploit the nesting technique further and either use it for other known mountain wave hot spots (e.g., Hoffmann et al., 2013, 2017) or longer periods. In addition, the northern hemisphere will be analysed in the future where PSC formation and ozone depletion has been shown to be highly sensitive to the existence of mountain waves (e.g., Tabazadeh et al., 2000; Eckermann et al., 2006; Dörnbrack et al., 2012; Khosrawi et al., 2018).

*Code and data availability.* Licences of the ICON code are currently managed by the Max-Planck-Institute for Meteorology (MPI-M) and the German Weather Service (DWD). Please visit https://code.mpimet.mpg.de/projects/iconpublic/wiki/How_to_obtain_the_model_code (last access on 05 November 2020) for further information. For ART, please contact Bernhard Vogel (bernhard.vogel@kit.edu). The version 2 data of CALIOP PSCs first published in Pitts et al. (2018) can be directly obtained by contacting Michael Pitts (michael.c.pitts@nasa.gov) and will be made available soon at CALIPSO Science Team (2015). The AIRS data is distributed by the NASA Goddard Earth Sci-

ences Data Information and Services Center (AIRS project, 2007). The AIRS gravity wave data sets used in this study can be accessed at https://datapub.fz-juelich.de/slcs/airs/gravity_waves (last access on 05 November 2020). The code to transfer model PSC data into the radiation space of CALIOP has been recently published as supplement by Steiner et al. (2020).





## Appendix A: Reactions and their rate constants

In this appendix, we provide all chemical reactions that are used in this study. Table A1 shows the 93 gasphase reactions. The gasphase reaction rate constants are computed as part of the MECCA module. Table A2 shows the 11 heterogeneous reactions where the reaction rate constants are calculated by the PSC scheme of ICON-ART. Finally, the 38 photolytic reactions are summarised in Table A3. Photolysis rates are calculated by the CloudJ module (Prather, 2015).

**Table A1.** Gasphase reactions. The reaction rates constants in the second column are given in units of either $\mathrm{cm^3 s^{-1}}$ or $\mathrm{cm^6 s^{-2}}$ depending on the type of the reaction. Reaction rate constants are provided by the MECCA module, incorporated in ICON-ART.

| Reaction | Rate Constant (cgs-system) |
|---|---|
| $O_2 + O(^1D) \rightarrow O(^3P) + O_2$ | `3.3E-11*EXP(55./temp)` |
| $O_2 + O(^3P) + M \rightarrow O_3 + M$ | `6.E-34*((temp/300.)**(-2.4))*cair` |
| $O_3 + O(^1D) \rightarrow 2\,O_2$ | `1.2E-10` |
| $O_3 + O(^3P) \rightarrow 2\,O_2$ | `8.E-12*EXP(-2060./temp)` |
| $H + O_2 + M \rightarrow HO_2 + M$ | `k_3rd(temp,cair,4.4E-32,1.3,7.5E-11,-0.2,0.6)` |
| $H + O_3 \rightarrow OH + O_2$ | `1.4E-10*EXP(-470./temp)` |
| $H_2 + O(^1D) \rightarrow H + OH$ | `1.2E-10` |
| $OH + O(^3P) \rightarrow H + O_2$ | `1.8E-11*EXP(180./temp)` |
| $OH + O_3 \rightarrow HO_2 + O_2$ | `1.7E-12*EXP(-940./temp)` |
| $OH + H_2 \rightarrow H_2O + H$ | `2.8E-12*EXP(-1800./temp)` |
| $HO_2 + O(^3P) \rightarrow OH + O_2$ | `3.E-11*EXP(200./temp)` |
| $HO_2 + O_3 \rightarrow OH + 2\,O_2$ | `1.E-14*EXP(-490./temp)` |
| $HO_2 + H \rightarrow 2\,OH$ | `7.2E-11` |
| $HO_2 + H \rightarrow H_2 + O_2$ | `6.9E-12` |
| $HO_2 + H \rightarrow O(^3P) + H_2O$ | `1.6E-12` |
| $HO_2 + OH \rightarrow H_2O + O_2$ | `4.8E-11*EXP(250./temp)` |
| $HO_2 + HO_2 + M \rightarrow H_2O_2 + O_2 + M$ | `k_HO2_HO2` |
| $H_2O + O(^1D) \rightarrow 2\,OH$ | `1.63E-10*EXP(60./temp)` |
| $H_2O_2 + OH \rightarrow H_2O + HO_2$ | `1.8E-12` |
| $N + O_2 \rightarrow NO + O(^3P)$ | `1.5E-11*EXP(-3600./temp)` |
| $N_2 + O(^1D) \rightarrow O(^3P) + N_2$ | `2.15E-11*EXP(110./temp)` |
| $N_2O + O(^1D) \rightarrow 2\,NO$ | `7.25E-11*EXP(20./temp)` |
| $N_2O + O(^1D) \rightarrow N_2 + O_2$ | `4.63E-11*EXP(20./temp)` |
| $NO + O_3 \rightarrow NO_2 + O_2$ | `3.E-12*EXP(-1500./temp)` |
| $NO + N \rightarrow O(^3P) + N_2$ | `2.1E-11*EXP(100./temp)` |





| Reaction | Rate Constant (cgs-system) |
|---|---|
| $NO_2 + O(^3P) \rightarrow NO + O_2$ | `5.1E-12*EXP(210./temp)` |
| $NO_2 + O_3 \rightarrow NO_3 + O_2$ | `1.2E-13*EXP(-2450./temp)` |
| $NO_2 + N \rightarrow N_2O + O(^3P)$ | `5.8E-12*EXP(220./temp)` |
| $NO_3 + NO \rightarrow 2\,NO_2$ | `1.5E-11*EXP(170./temp)` |
| $NO_3 + NO_2 + M \rightarrow N_2O_5 + M$ | `k_NO3_NO2` |
| $N_2O_5 + M \rightarrow NO_2 + NO_3 + M$ | `k_NO3_NO2/(2.7E-27*EXP(11000./temp))` |
| $NO + HO_2 \rightarrow NO_2 + OH$ | `3.3E-12*EXP(270./temp)` |
| $NO_2 + OH + M \rightarrow HNO_3 + M$ | `k_3rd(temp,cair,1.8E-30,3.0,2.8E-11,0.,0.6)` |
| $NO_2 + HO_2 + M \rightarrow HNO_4 + M$ | `k_NO2_HO2` |
| $HNO_3 + OH \rightarrow H_2O + NO_3$ | `k_HNO3_OH` |
| $HNO_4 + M \rightarrow NO_2 + HO_2 + M$ | `k_NO2_HO2/(2.1E-27*EXP(10900./temp))` |
| $HNO_4 + OH \rightarrow NO_2 + H_2O$ | `1.3E-12*EXP(380./temp)` |
| $CH_4 + O(^1D) \rightarrow .75\,CH_3O_2 + .75\,OH +$ $.25\,HCHO + .4\,H + .05\,H_2$ | `1.75E-10` |
| $CH_4 + OH \rightarrow CH_3O_2 + H_2O$ | `1.85E-20*EXP(2.82*log(temp)-987./temp)` |
| $CH_3O_2 + HO_2 \rightarrow CH_3OOH + O_2$ | `4.1E-13*EXP(750./temp)` |
| $CH_3O_2 + NO \rightarrow HCHO + NO_2 + HO_2$ | `2.8E-12*EXP(300./temp)` |
| $CH_3O_2 \rightarrow HCHO + HO_2$ | `2.*RO2*9.5E-14*EXP(390./temp)/` `(1.+1./26.2*EXP(1130./temp))` |
| $CH_3O_2 \rightarrow .5\,HCHO + .5\,CH_3OH + .5\,O_2$ | `2.*RO2*9.5E-14*EXP(390./temp)/(1.+26.2*EXP(-1130./temp))` |
| $CH_3OOH + OH \rightarrow .7\,CH_3O_2 + .3\,HCHO +$ $.3\,OH + H_2O$ | `k_CH2OOH_OH` |
| $HCHO + OH \rightarrow CO + H_2O + HO_2$ | `9.52E-18*EXP(2.03*log(temp)+636./temp)` |
| $CO + OH \rightarrow H + CO_2$ | `(1.57E-13+cair*3.54E-33)` |
| $Cl + O_3 \rightarrow ClO + O_2$ | `2.8E-11*EXP(-250./temp)` |
| $ClO + O(^3P) \rightarrow Cl + O_2$ | `2.5E-11*EXP(110./temp)` |
| $ClO + ClO \rightarrow Cl_2 + O_2$ | `1.0E-12*EXP(-1590./temp)` |
| $ClO + ClO \rightarrow 2\,Cl + O_2$ | `3.0E-11*EXP(-2450./temp)` |
| $ClO + ClO \rightarrow Cl + OClO$ | `3.5E-13*EXP(-1370./temp)` |
| $ClO + ClO + M \rightarrow Cl_2O_2 + M$ | `k_ClO_ClO` |
| $Cl_2O_2 + M \rightarrow ClO + ClO + M$ | `k_ClO_ClO/(1.72E-27*EXP(8649./temp))` |
| $Cl + H_2 \rightarrow HCl + H$ | `3.9E-11*EXP(-2310./temp)` |
| $Cl + HO_2 \rightarrow HCl + O_2$ | `4.4E-11-7.5E-11*EXP(-620./temp)` |





| Reaction | Rate Constant (cgs-system) |
| --- | --- |
| $Cl + HO_2 \rightarrow ClO + OH$ | `7.5E-11*EXP(-620./temp)` |
| $Cl + H_2O_2 \rightarrow HCl + HO_2$ | `1.1E-11*EXP(-980./temp)` |
| $ClO + OH \rightarrow .94\,Cl + .94\,HO_2 + .06\,HCl + .06\,O_2$ | `7.3E-12*EXP(300./temp)` |
| $ClO + HO_2 \rightarrow HOCl + O_2$ | `2.2E-12*EXP(340./temp)` |
| $HCl + OH \rightarrow Cl + H_2O$ | `1.7E-12*EXP(-230./temp)` |
| $HOCl + OH \rightarrow ClO + H_2O$ | `3.0E-12*EXP(-500./temp)` |
| $ClO + NO \rightarrow NO_2 + Cl$ | `6.2E-12*EXP(295./temp)` |
| $ClO + NO_2 + M \rightarrow ClNO_3 + M$ | `k_3rd_iupac(temp,cair,1.6E-31,3.4,7.E-11,0.,0.4)` |
| $ClNO_3 + O(^3P) \rightarrow ClO + NO_3$ | `4.5E-12*EXP(-900./temp)` |
| $ClNO_3 + Cl \rightarrow Cl_2 + NO_3$ | `6.2E-12*EXP(145./temp)` |
| $Cl + CH_4 \rightarrow HCl + CH_3O_2$ | `6.6E-12*EXP(-1240./temp)` |
| $Cl + HCHO \rightarrow HCl + CO + HO_2$ | `8.1E-11*EXP(-34./temp)` |
| $Cl + CH_3OOH \rightarrow HCHO + HCl + OH$ | `5.9E-11` |
| $ClO + CH_3O_2 \rightarrow HO_2 + Cl + HCHO$ | `3.3E-12*EXP(-115./temp)` |
| $CCl_4 + O(^1D) \rightarrow ClO + 3\,Cl$ | `3.3E-10` |
| $CH_3Cl + O(^1D) \rightarrow OH + Cl$ | `1.65E-10` |
| $CH_3Cl + OH \rightarrow H_2O + Cl$ | `2.4E-12*EXP(-1250./temp)` |
| $CH_3CCl_3 + O(^1D) \rightarrow OH + 3\,Cl$ | `3.E-10` |
| $CH_3CCl_3 + OH \rightarrow H_2O + 3\,Cl$ | `1.64E-12*EXP(-1520./temp)` |
| $CF_2Cl_2 + O(^1D) \rightarrow ClO + Cl$ | `1.4E-10` |
| $CFCl_3 + O(^1D) \rightarrow ClO + 2\,Cl$ | `2.3E-10` |
| $Br + O_3 \rightarrow BrO + O_2$ | `1.7E-11*EXP(-800./temp)` |
| $BrO + O(^3P) \rightarrow Br + O_2$ | `1.9E-11*EXP(230./temp)` |
| $BrO + BrO \rightarrow 2\,Br + O_2$ | `2.7E-12` |
| $BrO + BrO \rightarrow Br_2 + O_2$ | `2.9E-14*EXP(840./temp)` |
| $Br + HO_2 \rightarrow HBr + O_2$ | `7.7E-12*EXP(-450./temp)` |
| $BrO + HO_2 \rightarrow HOBr + O_2$ | `4.5E-12*EXP(500./temp)` |
| $HBr + OH \rightarrow Br + H_2O$ | `6.7E-12*EXP(155./temp)` |
| $HOBr + O(^3P) \rightarrow OH + BrO$ | `1.2E-10*EXP(-430./temp)` |
| $Br_2 + OH \rightarrow HOBr + Br$ | `2.0E-11*EXP(240./temp)` |
| $BrO + NO \rightarrow Br + NO_2$ | `8.7E-12*EXP(260./temp)` |





| Reaction | Rate Constant (cgs-system) |
|---|---|
| $BrO + NO_2 + M \rightarrow BrNO_3 + M$ | `k_BrO_NO2` |
| $Br + HCHO \rightarrow HBr + CO + HO_2$ | `7.7E-12*EXP(-580./temp)` |
| $CH_3Br + OH \rightarrow H_2O + Br$ | `2.35E-12*EXP(-1300./temp)` |
| $BrO + ClO \rightarrow Br + OClO$ | `1.6E-12*EXP(430./temp)` |
| $BrO + ClO \rightarrow Br + Cl + O_2$ | `2.9E-12*EXP(220./temp)` |
| $BrO + ClO \rightarrow BrCl + O_2$ | `5.8E-13*EXP(170./temp)` |
| $SO_2 + OH \rightarrow H_2SO_4 + HO_2$ | `k_3rd(temp,cair,3.3E-31,4.3,1.6E-12,0.,0.6)` |





**Table A2.** Heterogeneous reactions. Reaction rate constants are calculated by the PSC scheme of ICON-ART. The uptake coefficients of the different PSC types are taken from Sander et al. (2011b), where not stated differently.

| Reaction | $\gamma_{STS}$ | $\gamma_{NAT}$ | $\gamma_{ice}$ |
|---|---|---|---|
| $N_2O_5 + H_2O \rightarrow 2\ HNO_3$ | $(a)$ | 0.0004 | 0.027 |
| $HOCl + HCl \rightarrow Cl_2 + H_2O$ | $(a)$ | 0.1 | 0.2 |
| $ClNO_3 + HCl \rightarrow Cl_2 + HNO_3$ | $(a)$ | 0.2 | 0.3 |
| $ClNO_3 + H_2O \rightarrow HOCl + HNO_3$ | $(a)$ | 0.004 | 0.3 |
| $N_2O_5 + HCl \rightarrow ClNO_2 + HNO_3$ | $(a)$ | 0.003 | 0.03 |
| $HOBr + HBr \rightarrow Br_2 + H_2O$ | $(a)$ | $0.1^{(b)}$ | 0.1 |
| $BrNO_3 + H_2O \rightarrow HOBr + HNO_3$ | $(a)$ | $0.001^{(b)}$ | 0.26 |
| $ClNO_3 + HBr \rightarrow BrCl + HNO_3$ | $(a)$ | 0.3 | 0.3 |
| $BrNO_3 + HCl \rightarrow BrCl + HNO_3$ | $(a)$ | $0.3^{(b)}$ | 0.26 |
| $HOCl + HBr \rightarrow BrCl + H_2O$ | $(a)$ | $0.3^{(b)}$ | $0.3^{(c)}$ |
| $HOBr + HCl \rightarrow BrCl + H_2O$ | $(a)$ | $0.1^{(b)}$ | 0.3 |

$(a)$ Parametrised according to Carslaw et al. (1995).

$(b)$ In analogy to similar reactions, coming from Carslaw et al. (1995)

$(c)$ In analogy to $HOBr + HCl$





**Table A3.** Photolytic reactions. Photolysis rates are calculated by the CloudJ module, incorporated in ICON-ART.

| Reaction |
| --- |
| $O_2 + h\nu \rightarrow O(^3P) + O(^3P)$ |
| $O_3 + h\nu \rightarrow O(^1D) + O_2$ |
| $O_3 + h\nu \rightarrow O(^3P) + O_2$ |
| $H_2O + h\nu \rightarrow H + OH$ |
| $H_2O_2 + h\nu \rightarrow 2\,OH$ |
| $N_2O + h\nu \rightarrow O(^1D) + N_2$ |
| $NO_2 + h\nu \rightarrow NO + O(^3P)$ |
| $NO + h\nu \rightarrow N + O(^3P)$ |
| $NO_3 + h\nu \rightarrow NO_2 + O(^3P)$ |
| $NO_3 + h\nu \rightarrow NO + O_2$ |
| $N_2O_5 + h\nu \rightarrow NO_2 + NO_3$ |
| $HNO_3 + h\nu \rightarrow NO_2 + OH$ |
| $HNO_4 + h\nu \rightarrow .667\,NO_2 + .667\,HO_2 + .333\,NO_3 + .333\,OH$ |
| $CH_3OOH + h\nu \rightarrow HCHO + OH + HO_2$ |
| $HCHO + h\nu \rightarrow H_2 + CO$ |
| $HCHO + h\nu \rightarrow H + CO + HO_2$ |
| $CO_2 + h\nu \rightarrow CO + O(^3P)$ |
| $CH_4 + h\nu \rightarrow CO + 0.31\,H + 0.69\,H_2 + 1.155\,H_2O$ |
| $Cl_2 + h\nu \rightarrow Cl + Cl$ |
| $Cl_2O_2 + h\nu \rightarrow 2\,Cl$ |
| $OClO + h\nu \rightarrow ClO + O(^3P)$ |
| $HOCl + h\nu \rightarrow OH + Cl$ |
| $ClNO_2 + h\nu \rightarrow Cl + NO_2$ |
| $ClNO_3 + h\nu \rightarrow Cl + NO_3$ |
| $ClNO_3 + h\nu \rightarrow ClO + NO_2$ |
| $CH_3Cl + h\nu \rightarrow Cl + CH_3O_2$ |
| $CCl_4 + h\nu \rightarrow 4\,Cl$ |
| $CH_3CCl_3 + h\nu \rightarrow 3\,Cl$ |
| $CFCl_3 + h\nu \rightarrow 3\,Cl$ |
| $CF_2Cl_2 + h\nu \rightarrow 2\,Cl$ |
| $Br_2 + h\nu \rightarrow Br + Br$ |
| $BrO + h\nu \rightarrow Br + O(^3P)$ |





| Reaction |
| --- |
| $HOBr + h\nu \rightarrow Br + OH$ |
| $BrNO_3 + h\nu \rightarrow 0.85\ Br + 0.85\ NO_3 + 0.15\ BrO + 0.15\ NO_2$ |
| $CH_3Br + h\nu \rightarrow Br + CH_3O_2$ |
| $CF_3Br + h\nu \rightarrow Br$ |
| $BrCl + h\nu \rightarrow Br + Cl$ |
| $CF_2ClBr + h\nu \rightarrow Br + Cl$ |

*Author contributions.* This paper is part of MW's thesis supervised by PB and RR. MW developed the module for PSCs in ICON-ART and performed the simulations with contributions by JB, OK, RR and PB. LH performed the comparison between AIRS and ICON-ART. BL, IT and MS helped to perform the comparison between ICON-ART and CALIOP. MW prepared the manuscript with contributions by all authors.

*Competing interests.* The authors declare that they have no conflict of interest.

*Acknowledgements.* Parts of the simulations were performed on bwUniCluster for which the authors acknowledge support by the state of Baden-Württemberg through bwHPC. Other parts of the work as well as evaluations were performed on the supercomputer ForHLR funded by the Ministry of Science, Research and the Arts Baden-Württemberg and by the Federal Ministry of Education and Research. This work was performed with the help of the Large Scale Data Facility at the Karlsruhe Institute of Technology funded by the Ministry of Science, Research and the Arts Baden-Württemberg and by the Federal Ministry of Education and Research. We acknowledge ECCAD for archiving and distributing the emission data. We acknowledge funding from the Initiative and Networking Fund of the Helmholtz Association through the projects "Digital Earth" and "Advanced Earth System Modelling Capacity". I. Tritscher was funded by the Deutsche Forschungsgemeinschaft (DFG) under project number 310479827.



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
