# Peer review of "Mountain-wave-induced polar stratospheric clouds and their representation in the global chemistry model ICON-ART"

_Atmospheric Chemistry and Physics, 2020_

## Referee Comment (RC1) · Anonymous Referee #1 · 9 Dec 2020

Weimer et al. investigate the polar stratospheric cloud (PSC) formation during an Antarctic mountain wave event using the ICON-ART model. The model applies local grid refinements (nesting) with two-way interaction to simulate the temperature fluctuation and wind shear during the mountain wave event. The model has a detailed PSC parameterization of three types of PSCs. The results are compared with CALIPSO and ARIS satellite data for PSC classifications and brightness temperature for validation. The manuscript also analyzes the impact of PSCs on ozone-related chemicals with and without mountain waves. Although more work needs to be done in the future regarding the model validations, this two-way nesting approach provides a new and useful pathway for future PSC and stratospheric ozone studies, because the localized

[Figure]

GW PSC formations usually cannot be treated in the global model properly.

I have four major concerns that I recommend further discussions in your manuscript.

First, your PSC parameterizations limit the number concentration to 2.3e-4 cm-3 for NAT and 0.25 cm-3 for ice. There's nothing wrong to have the assumptions since this is a PSC parameterization model, but I'm not convinced by your descriptions of the number density choices for NAT and ice. For NAT particles, the GW likely booster the NAT particle concentration and significantly increasing the surface area density of NAT. Do you have any evidence that your number concentration assumption is suitable for GW conditions? You refer to Fahey et al. 2001 for this concentration choice for all your NAT particles as small as 0.1 um. However, Fahey et al. mentioned a two-mode particle concentration. The 2.3e-4 cm-3 is only for particles larger than 10 $\mu$m. And particles $\sim$3 $\mu$m has a larger number concentration $\sim$1e-3 cm-3. I don't think it's proper to quote part of their measurements and ignore others. For ice concentration, in line 350, you said the mountain wave can increase the concentration to the order of a few cm-3 but your assumptions are smaller than that. In contrast, in line 404-405, you said "ice number concentration is set to the tropospheric value of 0.25 cm−3 which is too large in comparison to measurements". So, is 0.25 too high or too low? Could you explain it more?

The second question is about ozone chemistry (section 5.4.3 and 5.4.4). The temperature fluctuation causes a regional low temperature of 10 degrees than the average value. Have you considered the impact of the low temperature on the chemistry reaction rate? The uptake coefficients of chorine activation reactions are very sensitive to the temperature. GW probably has an even bigger impact on chlorine activation and ozone depletion than the enhanced surface area density (SAD) provided by PSCs. If your model setting has already considered the temperature fluctuation in the chemistry module, the impact on heterogeneous chemistry is not only from PSCs SAD but also from the temperature. Line 440:" At altitudes around 26 km, the additional ice PSCs activate both ClONO2 and HCl in the lee of the mountains." But your ice PSC locations

(Figure 13c) and Cl species (Figure 13e, 13f) locations are not correlated.

Question3: Section 5.2: Could you please explain why your model forms PSCs at higher temperatures than observed (all the three figures, Figure 5, 6, 7)? Is this because your model did not denitrify or dehydrate properly before July? There are several places related to this problem: Line 327: What is "this bin" referring to. If you are referring to temperature bins 184K and 188K. I think your model is overestimated and it is not negligible. Conclusions for section 5.2 (near Line 353). In addition to "some differences in NAT at low temperatures and the "Wave-ice" category", the simulation also has the problem that forms PSCs at high temperatures. Line 497-498: "The comparison with all CALIOP measurements within the Antarctic Peninsula nest demonstrated that the general formation of most of the PSC types in ICON-ART is similar with respect to temperature." I cannot agree with this statement since you form the STS and NAT at these higher temperatures that are not in CALIPSO.

Question4: Figure 2: why do you do the free-running in the third step (i.e. July 19 – July 29)? Is it because the generation of temperature fluctuation from gravity wave needs a free-running model? As you mentioned in many places in your manuscript that you cannot directly compare with the observations (like CALIPSO and AIRS) because of the free-running. Why don't you do a nudged run instead?

General comments:

Line 44-46: Is the +-15K near the altitude of PSC formation or for higher altitude. If it is for higher altitudes, it is not related to PSC formation or ozone depletion.

Line 47-49: This sentence is confusing. Are you talking about Arctic denitrification is closely connected to the heterogeneous nucleation of NAT on meteoric dust and NAT formed in mountain wave activity? How about other NAT formation pathways by previous studies, like homogeneous nucleation [Tabazadeh et al., 2002]? If NAT forms in the mountain wave, it still forms through a microphysical process like nucleation. Is it the heterogeneous nucleation of NAT on meteoric dust or it could be other processes?

Please rephrase the sentence.

Line 60: what's "the effect"? Are you passing the temperature fluctuation to the PSC formation module? Or you also pass other variables?

Line 92-93: What are these "several nucleation processes" referring to? Do you have more than one nucleation pathway for ice formation?

Line 95: Zhu et al., 2015. This approach is originally described in a WACCM paper by Wegner et al., 2013. Please cite Wegner et al. 2013 instead.

Figure 1: the y axis is not particle number concentration since it has um-1. Is it dNdlnr?

Line 304: how about the refractive indices of STS?

Line 305: These are not boundaries between STS, NAT, and ice. These are STS, NAT mixtures, and ice mixtures since particles are internally mixed. Also, for your simulated data, are you considering a mixture of different types also? Please specify here.

Line 306: I thought you calculate the R and beta but not the threshold. The dynamic threshold is determined by the denitrification and dehydration status. I don't think you need to calculate the threshold when it's a fixed boundary (for example, the STS category).

Line 308: Are you adding the uncertainties to the threshold or adding the uncertainties to the backscatter coefficients and backscatter ratio?

Line 325: what do you mean by "The development of pure STS particles is similar"? What does "development" mean?

Line 326: It seems the figure shows ICON-ART "overestimate" the STS at higher temperatures compared with CALIPSO, not "underestimate".

Line 332: "XH2O = 5 ppmv and XHNO3 = 10 ppbv" These values feel like before denitrification and dehydration (early winter), not in July. Are you sure you've considered

denitrification and dehydration here?

Line 350: please mention this value (0.25 cm-3) near Eqn (1).

Line 392: "where also H2SO4 is enhanced". Why H2SO4 is enhanced? Do you mean aerosols or gas? If you are using a prescribed H2SO4, why does the H2SO4 increase when you do the nesting?

Line 395: "In contrast to the literature, the NAT volume concentration decreases when the air masses approach the mountain wave." Is this sentence referring to your simulation or some other articles?

Figure 11: You don't have sulfate aerosol in the model? H2SO4 are all in the gas phase? If you don't have sulfate aerosol, it might be a problem for ozone chemistry since sulfate provides surface area too.

Line 417: "Due to the missing sink of H2SO4 by sedimentation of aerosols (see Sect. 3), the mixing ratio accumulates in the nested domains (two right columns)." Why the missing sink only affects the nested domain, not the global case.

Line 497-498: Again, if you compared with CALIPSO, your model output is not simply STS, NAT, and ice. They are all mixtures that fall into different PSC classification categories. You may want to check the wording through the whole content.

Appendix A: some reaction rates are not listed i.e. "k_HO2_HO2".

---

## Referee Comment (RC2) · Michael Weimer et al. · 24 Feb 2021

This is an interesting, timely and novel study, examining the ability of the ICON-ART chemistry-climate model with its flexible grid to resolve mountain waves over the Antarctic Peninsula and examine the impacts of this on PSC amounts and chemistry. The strength of this work is that it examines the impacts (of the inclusion of mountain wave-induced temperature fluctuations) on PSC chemistry, which has never really been done in such detail before. Consequently, the results/figures are very interesting, well thought out, and convincing - with a lot of time having obviously been spent preparing them. However, the interesting results are rather let down by the quality of the writing, which is in places is rather difficult to follow and unconvincing, and just not nearly polished/good enough. This is detailed below, but fundamentally some rigorous editing is

Interactive
comment

needed to address this - and to shorten the paper, as it feels over long. Additionally, the outstanding results were when the global model results were compared with the regional/40km results, but this comparison was not always made. I think the study would be much stronger if this deficiency was addressed. Addressing these suggestions (and the comments below) should be very achievable by the authors, and I hope will result in a much stronger and more impactful paper, that the novelty and importance of the results merits. In summary, this paper includes some very good results that will be of strong interest to ACP readers, but at the moment it will require major revisions before it is acceptable to be considered for publication.

Major comments

+ As it stands the paper is rather awkward to read and rather disjointed and would really benefit from some quite rigorous editing to make it more concise. I have given many examples below, but they are far from exhaustive. For example, at times the text is much too 'wordy', making it difficult to follow and unconvincing (see e.g. Section 3). The description of the results could also be tighter and more concise. The various sections also aren't well linked and the reader is never sure of the scope of the work. These issues really affect the readability/coherency of the paper. If you properly address these issues the paper will be much stronger and convincing and polished.

+ The Introduction seems rather limited and patchy, and not as coherent as it could be. I've given some suggestions as to how it could be improved below. But an obvious one would be to expand on why the Antarctic Peninsula has been identified as the focus for this case study, as well as more mention of other orographic hotspots in both the southern and northern hemispheres to reinforce the importance/impact of this work. Currently, we get to the final paragraph of the Introduction with little idea of what the 'description of the simulation' is, what model results are examined (PSC chemistry, GWs, dynamics?), or what satellite data is used in the study. I understand that this is elaborated on in the later sections, but these items also need to 'introduced' so that the paper is coherent, stronger, and easier to read. See also the comment below on lines

**252-255, which is the sort of information that should be included here.**

+ Line #54-61: I think that more information is required in this paragraph, especially on the difference/increase in resolution between the global domain and the refined grid in ICON-ART, and whether the refined grid is therefore suitable to resolve the relatively small horizontal scale mountain waves that typically form over the narrow Antarctic Peninsula. For example, Noel and Pitts (2012) used a resolution of 20 km to resolve these waves, while studies such as Orr et al. (2015) suggested that a much higher resolution of ∼4 km was required. Perhaps the resolution used in the refined grid is justified as suitable for longer climatological runs (lower computational cost), which requires a compromise. For example, line #60 claims that these waves will be 'directly simulated', but this doesn't necessarily mean that they are realistically represented. These sort of details would be helpful.

Noel, V. and Pitts, M.: Gravity wave events from mesoscale simulations, compared to polar stratospheric clouds observed from spaceborne lidar over the Antarctic Peninsula, J. Geophys. Res.,117, D11207, doi:10.1029/2011JD017318, 2012.

+ Section 2: This is a very lengthy and detailed description of the ICON-ART model, and especially its PSC scheme. Its not clear to me why such a detailed description is required (in the main text), as this has surely been explained elsewhere. To a non-chemist such a long section is difficult to follow. If such a detailed description is required, then explain in a sentence or so why. If such detail is not required, then please remove much of it. I would encourage that much of it is deleted or moved to the Appendix, as it made getting to the results difficult and frustrating as this had to be read firstly. I also don't see any point in all the equations in the Appendix being included, so please delete.

Additionally, the description of the PSC scheme needs to discuss how it deals with the mountain wave temperature perturbations. For example, if the 'warm phase' of the mountain wave temperatures results in temperatures being higher than the PSC formation threshold temperatures, then what happens? Do the PSCs evaporate instantaneously? Also, please describe whether the PSC scheme is able to advect PSCs downstream of the orography, as seen in observations (Eckermann et al., 2009).

+ Figure 5: Please include results from the global ICON-ART version in Figure 5 so that we can clearly see the benefit of moving from a resolution of 160 km to 40 km. The resulting plot would be very strong and highly novel, clearly demonstrating the direct benefits of resolving mountain waves in terms of PSCs for the first time (rather than indirectly, such as how often PSC formation temperature thresholds are exceeded, such as in Orr et al. 2020). Results for the global model should also be added to Figs 6 and 7.

Minor comments

+ Line #26. Please quantify the horizontal scale of mountain waves. Also, a high model resolution is required to resolve the actual wave dynamics/evolution, and not just the orography. For example, it is thought that as many as 8 grid points is required to adequately resolve a gravity wave.

+ Line #28: You have mentioned the resolution of global models, but not the resolution for mesoscale models. Please add this.

+ Line #30: I don't follow your statement that 'A method to bridge this gap for interactive calculations is missing so far', as in this paragraph you have stated that solutions such as parameterizing the effects of orographic GWs exist. Please revise.

+ Line #36: Some mention of the AIRS GW climatology (Hoffmann et al. 2016) would also be appropriate, as well as the various orographic hotspots that this reveals in both the southern and northern hemispheres.

+ Lines #52-52: The statement that Alexander et al. (2011) concluded that about 30% of all southern hemispheric PCSs can be related to mountain waves is incorrect. Alexander et al. (2011) includes the caveat that this number is only for the latitudinal

range 60-70S. A better study to cite is probably Alexander et al (2013), which states that 'For all types of PSC, 5% in the whole Antarctic and 12% in the whole Arctic are attributed to OGW forcing'. Please revise the manuscript to reflect this, and also check that the numbers quoted from the other studies are correct and consistent with Alexander et al. (2013).

Alexander, S. P., Klekociuk, A. R., McDonald, A. J., and Pitts, M. C. (2013), Quantifying the role of orographic gravity waves on polar stratospheric cloud occurrence in the Antarctic and the Arctic, J. Geophys. Res. Atmos., 118, 11,493– 11,507, doi:10.1002/2013JD020122.

+ Lines #77-78. Please revise the sentence '... used which is similar to other studies (e.g., Stone et al., 2019; Zambri et al., 2019; Nakajima et al.,2020) and can be found in Appendix A.' as its not clear. Also please elaborate why these equations need to be listed in the Appendix, as its not clear. If they are not necessary then please delete them to improve the flow/readability of the paper.

+ Line #87: I have made this point already, but you can't simply make statements such as 'simulate e.g. mountain waves' without justifying this, such as explaining the scale of the waves and the grid scales used by the model. You need to do this.

+ Equations (1) and (2): Please check that the temperature T is defined before it is first used. (It is defined in line #119, after these equations.)

+ Line #175: Out of the blue we are informed that 'In order to compare the results of the PSC scheme in ICON-ART with satellite measurements and to investigate the impact of the nesting technique on mountain-wave induced PSCs, ....'. As explained above, please make this clearer much earlier on. This would make the paper much clearer.

+ Line #179: Some basic information needs to be included such as to the synoptic conditions that result in the formation of the orographic wave. For example, presumably this is due to an easterly wind over the Antarctic Peninsula? Please include this sort of

information. Also, please put the period (July) examined into the wider context of the austral winter / PSC season.

+ Line #185: Has EMAC been defined?

+ Lines #191-194: These two sentences are unclear. Please revise.

+ Lines #204-209: This paragraph could be better written.

+ Table 2: The caption just seems like a repeat of the text in the section. Please consider revising this.

+ Line #207: Here you mention that a grid spacing of 40 km will be used for the nested simulation, but there is still no justification as to why this resolution was chosen and why it is thought to be appropriate. See other comments above.

+ Line #228: Please revise the sentence: '... on the one hand and exclude tropospheric clouds on the other hand'.

+ Lines #252-255: This is the sort of information that needs to have been included much earlier, say at the end of the Introduction. So that you are clearly explaining early on to the reader the scope of the paper.

+ Line #259: I think that the results are much more nuanced and subtle than simply saying 'can be directly simulated with the resolution of 40 km'. Especially, because you are not showing any evidence at this point that the 40 km simulation of the wave is realistic.

+ Lines #262-264: Please revise this text. As already mentioned above, there is more to accurately representing a mountain wave than just resolving the steepness/height of the orography. This is a rather naive understanding of the problem.

+ Figure 4: The 80 km results don't seem to be mentioned.

+ Line #265: Its not clear what you are referring to by 'Therefore, the flow over the
mountain range is under represented in the model'. The global model with a much smoother/lower orography might actually have enhanced flow over the Peninsula, although this is not a given as the model has a sub-grid scale orographic drag scheme to represent unresolved drag. Please revise.

+Lines #265-266: This is not clear. Please revise.

+ Line #275: As I have tried to emphasize, I think that you need to make these sort of statements more comparative, ie compared to the 160 km model, the 40 km model represents a well defined mountain wave with an amplitude of 10 K – this wave is entirely absent in the 160 km model. So you are trying to justify that you have produced a step change improvement in the representation of mountain waves in the model by going from 160 -> 40 km.

+ Line #280: Spelling. Fourth.

+ Line #314: Not clear what 'datasets' you are referring to here. Please clarify in the text.

+ Line #317: Its not at all clear what 'the results can be found in Fig. 5' are referring to. Maybe you are referring to the model v CALIOP PSC volume concentration, but this was introduced many paragraphs previously.

+ Figure 5: I'm not convinced that a statistical analysis is possible for such a short period considered. Please justify that this in the text, or amend the language so that you say that you are simply making a comparison for the period examined. For example, what is the frequency of CALIOP measurements etc.

+ Figure 5: Its not clear what the temperature values are on the figure. The ICON-ART panel has two different sets of values, which are unclear. The CALIOP panel has different values, which range from 23K to 349K (I think this is wrongly labelled). This is confusing, and makes it unclear how to compare the model results v CALIOP. Consequently, I found it difficult to follow the explanation of the Figure 5 in lines #324-

330.

+ Lines #324-330: You haven't referred to where the improvement resulting from resolving mountain waves would be expected in terms of PSCs. Presumably, you would expect more of a benefit for ice PSCs due to their lower temperature formation threshold compared to NAT and STS PSCs, ie they require the additionally cooling from GWs to exceed this temperature. Please clarify this when referring to Figure 5.

+ Lines #367-369: How important are these missing fine scale features? What are the implications for the simulated GW temperature perturbations? Does this suggest that ideally a higher resolution is required? Can you connect these deficiencies to the results in Figure 5, 6 and 7?

+ Figure 11: The 80 km results are included but not discussed. How do these compare with the 40 km results?

+ Figures 13: It's a little confusing jumping from the Peninsula region to a global domain. Perhaps this needs a separate sub-section or added to the section describing Figure 14.
* * *

---

## Author Comment (AC1) · 6 Apr 2021

**Response to Referee #1**

Dear Referee,

thank you for your detailed review of our manuscript. Please find our responses to your comments below.

Major changes include

- a comprehensive revision of the introduction

- moving the description of the PSC scheme in the appendix and keeping just the important information in the main text

- restructuring of the simulation description

- correcting an error in the CALIOP comparison and accordingly a revision of Sect. 5.2

- moving the reaction mechanism from the appendix to the supplementary material

Yours sincerely and on behalf of all co-authors,
Michael Weimer

**1 Major Comments**

1.1 First, your PSC parameterizations limit the number concentration to 2.3e-4 cm-3 for NAT and 0.25 cm-3 for ice. There's nothing wrong to have the assumptions since this is a PSC parameterization model, but I'm not convinced by your descriptions of the number density choices for NAT and ice. For NAT particles, the GW likely booster the NAT particle concentration and significantly increasing the surface area density of NAT. Do you have any evidence that your number concentration assumption is suitable for GW conditions? You refer to Fahey et al. 2001 for this concentration choice for all your NAT particles as small as 0.1 um. However, Fahey et al. mentioned a two-mode particle concentration. The 2.3e-4 cm-3 is only for particles larger than 10 um. And particles $\sim$ 3 um has a larger number concentration $\sim$ 1e-3 cm-3. I don't think it's proper to quote part of their measurements and ignore others. For ice concentration, in line 350, you said the mountain wave can increase the concentration to the order of a few cm-3 but your assumptions are smaller than that. In contrast, in line 404-405, you said "ice number concentration is set to the tropospheric value of 0.25 cm-3 which is too large in comparison to measurements". So, is 0.25 too high or too low? Could you explain it more?

Our NAT formation procedure bases on van den Broek et al. (2004). Therefore, we use the size distribution provided by van den Broek et al. (2004) who show that it leads to denitrification comparable to measurements. They also provide a quite comprehensive explanation why they are using this value. We now state this clearly and cite their paper.

For ice, the used particle number concentration of 0.25 cm-3 originates from the tropospheric hydrometeor microphysics of the model. We clarified the statement in line 405. The number (from a model tuned for global performance) is too low for regional

mountain waves.

1.2 The second question is about ozone chemistry (section 5.4.3 and 5.4.4). The temperature fluctuation causes a regional low temperature of 10 degrees than the average value. Have you considered the impact of the low temperature on the chemistry reaction rate? The uptake coefficients of chorine activation reactions are very sensitive to the temperature. GW probably has an even bigger impact on chlorine activation and ozone depletion than the enhanced surface area density (SAD) provided by PSCs. If your model setting has already considered the temperature fluctuation in the chemistry module, the impact on heterogeneous chemistry is not only from PSCs SAD but also from the temperature. Line 440: "At altitudes around 26 km, the additional ice PSCs activate both ClONO2 and HCl in the lee of the mountains." But your ice PSC locations (Figure 13c) and Cl species (Figure 13e, 13f) locations are not correlated.

Yes, the whole chemistry, including PSCs and photolysis, is called in all nests using the temperature as shown in the figures. We added statements regarding the temperature dependence to the sentences where only PSC SADs were mentioned as a source. In addition, we included in the simulation description that the chemistry is called in the nests as well.

As part of our response to referee 2, we included a description of the mountain wave event from a meteorological point of view. This includes that easterly winds over the Antarctic Peninsula lead to the mountain wave. Figure 13 c) shows that the ice PSC in the simulation with the nests reaches one kilometre higher than in the simulation without the nests. We replaced the panel by a figure with a refined colour bar. It shows better that the ice PSCs and low temperatures in the lee of the mountain reach higher in the simulation with the nests. In correspondence with the winds, chlorine is activated in the mountain wave and the activated chlorine is then transported downstream.

1.3 Question3: Section 5.2: Could you please explain why your model forms PSCs at higher temperatures than observed (all the three figures, Figure 5, 6, 7)? Is this because your model did not denitrify or dehydrate properly before July? There are several places related to this problem: Line 327: What is "this bin" referring to. If you are referring to temperature bins 184K and 188K. I think your model is overestimated and it is not negligible. Conclusions for section 5.2 (near Line 353). In addition to "some differences in NAT at low temperatures and the "Wave-ice" category", the simulation also has the problem that forms PSCs at high temperatures. Line 497-498: "The comparison with all CALIOP measurements within the Antarctic Peninsula nest demonstrated that the general formation of most of the PSC types in ICON-ART is similar with respect to temperature." I cannot agree with this statement since you form the STS and NAT at these higher temperatures that are not in CALIPSO.

We found an error in our analysis to derive optical properties from the modelled PSCs. We corrected this, included updated figures and revised the manuscript accordingly.

This correction clearly shows that no PSCs are formed at temperatures higher than TNAT.

1.4 Question4: Figure 2: why do you do the free-running in the third step (i.e. July 19 – July29)? Is it because the generation of temperature fluctuation from gravity wave needs a free-running model? As you mentioned in many places in your manuscript that you cannot directly compare with the observations (like CALIPSO and AIRS) because of the free-running. Why don't you do a nudged run instead?

We opted for a simpler approach to allow the model to track largely the general meteorological development, yet leave it as free as possible to develop the waves and the corresponding composition changes. The reason is that we are interested in the impact of the higher resolution domain's feedback on the global domain, which can only be done in a free-running system. A global nudging would interfere with this aim.

**2 Minor Comments**

2.1 Line 44-46: Is the +-15K near the altitude of PSC formation or for higher altitude. If it is for higher altitudes, it is not related to PSC formation or ozone depletion.

The range is taken from Carslaw et al. (1998b), now cited in this context, and are related to ozone depletion.

2.2 Line 47-49: This sentence is confusing. Are you talking about Arctic denitrification is closely connected to the heterogeneous nucleation of NAT on meteoric dust and NAT formed in mountain wave activity? How about other NAT formation pathways by previous studies, like homogeneous nucleation [Tabazadeh et al., 2002]? If NAT forms in the mountain wave, it still forms through a microphysical process like nucleation. Is it the heterogeneous nucleation of NAT on meteoric dust or it could be other processes? Please rephrase the sentence.

We removed the first part of the sentence and rephrased it.

2.3 Line 60: what's "the effect"? Are you passing the temperature fluctuation to the PSC formation module? Or you also pass other variables?

We added more information about how the two-way nesting is applied in the model in the introduction.

**2.4** Line 92-93: What are these "several nucleation processes" referring to? Do you have more than one nucleation pathway for ice formation?

We replaced this statement by "heterogeneous nucleation of cloud ice, nucleation of cloud ice due to homogeneous freezing of cloud water and depositional growth and sublimation of cloud ice". Further information can be found in the cited technical description of Doms et al. (2011).

**2.5** Line 95: Zhu et al., 2015. This approach is originally described in a WACCM paper by Wegner et al., 2013. Please cite Wegner et al. 2013 instead. Figure 1: the y axis is not particle number concentration since it has um-1. Is it dNdlnr?

We replaced the citation. The notion "particle number concentration" originates from the distributions shown by Figure 3 of van den Broek et al. (2004). We added $\Delta N/\Delta r$ to Fig. 1.

**2.6** Line 304: how about the refractive indices of STS?

The refractive index for STS is assumed to be 1.44 (Krieger et al., 2000). We included this in the manuscript.

Ulrich K. Krieger, Juliane C. Mössinger, Beiping Luo, Uwe Weers, and Thomas Peter, "Measurement of the refractive indices of H2SO4-HNO3-H2O solutions to stratospheric temperatures," Appl. Opt. 39, 3691-3703 (2000)

2.7   Line 305: These are not boundaries between STS, NAT, and ice. These are STS, NAT mixtures, and ice mixtures since particles are internally mixed. Also, for your simulated data, are you considering a mixture of different types also?  Please specify here.

We added "mixtures" to NAT and ice.  In the model, the PSC particles are externally mixed. These external mixtures lead to the optical properties ($R$ and $\beta$) for each grid point.

2.8   Line 306: I thought you calculate the R and beta but not the threshold.  The dynamic threshold is determined by the denitrification and dehydration status. I don't think you need to calculate the threshold when it's a fixed boundary (for example, the STS category).

As mentioned above, we had an error in the method and revised the whole paragraph.

2.9   Line 308: Are you adding the uncertainties to the threshold or adding the uncertainties to the backscatter coefficients and backscatter ratio?

The uncertainties are (1) added to the thresholds and (2) used as standard deviation for the normal distribution (now Eqs. 1 and 2) that is applied to introduce noise to the calculated optical properties.

2.10   Line 325: what do you mean by "The development of pure STS particles is similar"? What does "development" mean?

This sentence was removed as part of the complete revision of the section.

2.11 Line 326: It seems the figure shows ICON-ART "overestimate" the STS at higher temperatures compared with CALIPSO, not "underestimate".

We added more information about the temperature bins to make this clearer.

2.12 Line 332: "XH2O = 5 ppmv and XHNO3 = 10 ppbv" These values feel like before denitrification and dehydration (early winter), not in July. Are you sure you've considered denitrification and dehydration here?

These values are only used to have a reference temperature (TNAT / Tice) that can be applied to both CALIOP and ICON-ART and vary with pressure. We have clarified this in the text. In addition, we adapted the input values to 2.5 ppmv and 2 ppbv, based on MLS measurements shown in Tritscher et al. (2019).

2.13 Line 350: please mention this value (0.25 cm-3) near Eqn (1).

We added it below the equation (now Eq. A1).

2.14 Line 392: "where also H2SO4 is enhanced". Why H2SO4 is enhanced? Do you mean aerosols or gas? If you are using a prescribed H2SO4, why does the H2SO4 increase when you do the nesting?

This is a small artefact of the setup because H2SO4 is prescribed in the global domain, only. We added this to Sect. 3.

As pointed out in the description of the PSC scheme, the total gaseous H2SO4 is assumed to be in the aerosol. Therefore, the first row of Fig. 11 (now Fig. 12) shows the gaseous H2SO4 that is used to form binary aerosol/STS.

Since H2SO4 does not have a sink in the setup used, these enhancements are most probably a result of this missing sink and advection processes.

2.15    Line 395: "In contrast to the literature, the NAT volume concentration decreases when the air masses approach the mountain wave." Is this sentence referring to your simulation or some other articles? Figure 11: You don't have sulfate aerosol in the model? H2SO4 are all in the gasphase? If you don't have sulfate aerosol, it might be a problem for ozone chemistry since sulfate provides surface area too.

We included Carslaw et al., 1999 and Svendsen et al., 2005 as literature reference in the manuscript.

The module by Carslaw et al. (1995) accounts for binary aerosol, too, and the resulting particle volume concentration does not distinguish between sulfate and STS particles. Therefore, we changed all notions of STS to "liquid" in the manuscript and figures, where it is needed.

2.16    Line 417: "Due to the missing sink of H2SO4 by sedimentation of aerosols (see Sect.3), the mixing ratio accumulates in the nested domains (two right columns)." Why the missing sink only affects the nested domain, not the global case.

As mentioned above (your comment on Line 392), H2SO4 is prescribed only in the global domain.

2.17  Line 497-498: Again, if you compared with CALIPSO, your model output is not simply STS, NAT, and ice. They are all mixtures that fall into different PSC classification categories. You may want to check the wording through the whole content.

We replaced the notion "Type" by "Category" throughout the manuscript to distinguish between the NAT, STS and ice types of ICON-ART and the categories of CALIOP.

2.18  Appendix A: some reaction rates are not listed i.e. "k_HO2_HO2".

We included the reaction rate constants in the table (now Table S1 in the supplement) and also listed all abbreviations in the caption.

---

## Author Comment (AC2) · 6 Apr 2021

**Response to Referee #2**

Dear Referee,

thank you for your detailed review of our manuscript. Please find our responses to your comments below.
Major changes include

- a comprehensive revision of the introduction

- moving the description of the PSC scheme in the appendix and keeping just the important information in the main text

- restructuring of the simulation description

- correcting an error in the CALIOP comparison and accordingly a revision of Sect. 5.2

- moving the reaction mechanism from the appendix to the supplementary material

Yours sincerely and on behalf of all co-authors,
Michael Weimer

**1  Major comments**

1.1   As it stands the paper is rather awkward to read and rather disjointed and would really benefit from some quite rigorous editing to make it more concise. I have given many examples below, but they are far from exhaustive. For example, at times the text is much too 'wordy', making it difficult to follow and unconvincing (see e.g. Section 3). The description of the results could also be tighter and more concise. The various sections also aren't well linked and the reader is never sure of the scope of the work. These issues really affect the readability/coherency of the paper. If you properly address these issues the paper will be much stronger and convincing and polished.

To address your concerns, we have completely revised the Introduction and the section about the CALIOP comparison with ICON-ART. We moved the PSC scheme description in the appendix, thus making the main part of the manuscript more concise and focused on the results. We shortened and rearranged Sect. 3 which, we hope, improves the readability of the section. Of course, we also addressed the specific comments as well.

1.2   The Introduction seems rather limited and patchy, and not as coherent as it could be. I've given some suggestions as to how it could be improved below. But an obvious one would be to expand on why the Antarctic Peninsula has been identified as the focus for this case study, as well as more mention of other orographic hotspots in both the southern and northern hemispheres to reinforce the importance/impact of this work. Currently, we get to the final paragraph of the Introduction with little idea of what the 'description of the simulation' is, what model results are examined (PSC chemistry, GWs, dynamics?), or what satellite data is used in the study. I understand that this is elaborated on in the later sections, but these items also need to 'introduced' so that the paper is coherent, stronger, and easier to read. See also the comment below on lines #252-255, which is the sort of information that should be included here.

We completely revised the introduction in order to account for all the points you made here, and below.

1.3   Line #54-61: I think that more information is required in this paragraph, especially on the difference/increase in resolution between the global domain and the refined grid in ICON-ART, and whether the refined grid is therefore suitable to resolve the relatively small horizontal scale mountain waves that typically form over the narrow Antarctic Peninsula. For example, Noel and Pitts (2012) used a resolution of 20 km to resolve these waves, while studies such as Orr et al. (2015) suggested that a much higher resolution of $\sim$ 4 km was required. Perhaps the resolution used in the refined grid is justified as suitable for longer climatological runs (lower computational cost), which requires a compromise. For example, line #60 claims that these waves will be 'directly simulated', but this doesn't necessarily mean that they are realistically represented. These sort of details would be helpful.

We included a discussion of the resolutions in the introduction, which should provide the reader with a better overview how to interpret the 40 km resolution. In brief, it is our goal to start with a global resolution comparable to other CCMs and see the impact of the two-way nesting in the region of the Antarctic Peninsula when the large-scale flow can excite waves triggering PSCs.

1.4 Section 2: This is a very lengthy and detailed description of the ICON-ART model, and especially its PSC scheme. Its not clear to me why such a detailed description is required (in the main text), as this has surely been explained elsewhere. To a non-chemist such a long section is difficult to follow. If such a detailed description is required, then explain in a sentence or so why. If such detail is not required, then please remove much of it. I would encourage that much of it is deleted or moved to the Appendix, as it made getting to the results difficult and frustrating as this had to be read firstly. I also don't see any point in all the equations in the Appendix being included, so please delete. Additionally, the description of the PSC scheme needs to discuss how it deals with the mountain wave temperature perturbations. For example, if the 'warm phase' of the mountain wave temperatures results in temperatures being higher than the PSC formation threshold temperatures, then what happens? Do the PSCs evaporate instantaneously? Also, please describe whether the PSC scheme is able to advect PSCs downstream of the orography, as seen in observations (Eckermann et al., 2009).

We moved the main parts of the PSC scheme description to the appendix. Some statements are left in Sect. 2 because they are required for later sections. We also added a paragraph about the scheme's design with respect to mountain waves. Especially, the PSC scheme in ICON-ART is able to transport ice and NAT PSCs downstream as they are calculated prognostically.

From our point of view, the chemical reactions in the appendix are important because they follow the guideline of ACP: "2. A paper should contain sufficient detail and references to public sources of information to permit the author's peers to replicate the work." (see https://www.atmospheric-chemistry-and-physics.net/policies/obligations_for_authors.html, accessed on March 6, 2021). But we agree with your point that it is not needed in the main text and moved the equations to the supplement.

1.5   Figure 5: Please include results from the global ICON-ART version in Figure 5 so that we can clearly see the benefit of moving from a resolution of 160 km to 40 km. The resulting plot would be very strong and highly novel, clearly demonstrating the direct benefits of resolving mountain waves in terms of PSCs for the first time (rather than indirectly, such as how often PSC formation temperature thresholds are exceeded, such as in Orr et al. 2020). Results for the global model should also be added to Figs 6 and 7.

We included the new panels in Figs. 5-7. They clearly demonstrate the improvement using a higher resolution around the Antarctic Peninsula because temperatures lower than about 180 K don't occur in the resolution of 160 km. Since we found an error in our analysis comparing ICON-ART with CALIOP thanks to comments by referee 1, we comprehensively revised Sect. 5.2.

**2   Minor Comments**

2.1   + Line #26. Please quantify the horizontal scale of mountain waves. Also, a high model resolution is required to resolve the actual wave dynamics/evolution, and not just the orography. For example, it is thought that as many as 8 grid points is required to adequately resolve a gravity wave.

We included the horizontal scale of orographic wavelength and the requirement to the number of grid points in the introduction.

2.2  + Line #28: You have mentioned the resolution of global models, but not the resolution for mesoscale models. Please add this.

We included some examples of the resolution of mesoscale models used in the past for detection of mountain-wave induced PSCs.

2.3  + Line #30: I don't follow your statement that 'A method to bridge this gap for interactive calculations is missing so far', as in this paragraph you have stated that solutions such as parameterizing the effects of orographic GWs exist. Please revise.

We changed the statement to "An approach for global CCMs to directly benefit from high-resolution simulations is missing so far."

2.4  + Line #36: Some mention of the AIRS GW climatology (Hoffmann et al. 2016) would also be appropriate, as well as the various orographic hotspots that this reveals in both the southern and northern hemispheres.

We mentioned other hotspots of mountain-wave induced PSCs. We also included Hoffmann et al. (2016).

2.5  + Lines #52-52: The statement that Alexander et al. (2011) concluded that about 30% of all southern hemispheric PCSs can be related to mountain waves is incorrect. Alexander et al. (2011) includes the caveat that this number is only for the latitudinal range 60-70S. A better study to cite is probably Alexander et al (2013), which states that 'For all types of PSC, 5% in the whole Antarctic and 12% in the whole Arctic are attributed to OGW forcing'. Please revise the manuscript to reflect this, and also check that the numbers quoted from the other studies are correct and consistent with Alexander et al. (2013).

We replaced the statement by Alexander et al. (2013).

2.6  + Lines #77-78. Please revise the sentence '...used which is similar to other studies (e.g., Stone et al., 2019; Zambri et al., 2019; Nakajima et al.,2020) and can be found in Appendix A.' as its not clear. Also please elaborate why these equations need to be listed in the Appendix, as its not clear. If they are not necessary then please delete them to improve the flow/readability of the paper.

We separated the sentences and included a sentence listing the chemical families covered by the reaction system.

2.7  + Line #87: I have made this point already, but you can't simply make statements such as 'simulate e.g. mountain waves' without justifying this, such as explaining the scale of the waves and the grid scales used by the model. You need to do this.

We replaced the sentence by "Thus, the global domain is nudged towards the values in the nests, which will be further investigated in Sect. 5."

2.8 + Equations (1) and (2): Please check that the temperature T is defined before it is first used. (It is defined in line #119, after these equations.)

We added all symbols for equations 1 and 2 in the same sentence (now in the appendix Eqs. A1 and A2).

2.9 + Line #175: Out of the blue we are informed that 'In order to compare the results of the PSC scheme in ICON-ART with satellite measurements and to investigate the impact of the nesting technique on mountain-wave induced PSCs,....'. As explained above, please make this clearer much earlier on. This would make the paper much clearer.

We have included this in the introduction.

2.10 + Line #179: Some basic information needs to be included such as to the synoptic conditions that result in the formation of the orographic wave. For example, presumably this is due to an easterly wind over the Antarctic Peninsula? Please include this sort of information. Also, please put the period (July) examined into the wider context of the austral winter / PSC season.

We added a figure to the manuscript showing the meteorological conditions at the mountain wave event and put the July in the context of the ozone year 2008.

2.11 + Line #185: Has EMAC been defined?

Yes, on line 138. And also in the revised manuscript in Sect. 2.

2.12  + Lines #191-194: These two sentences are unclear. Please revise.

We explained the reason why we are using emissions only by CFC-11 in a clearer way.

2.13  + Lines #204-209: This paragraph could be better written.

We separated the reasoning and the description of the third part of the simulation.

2.14  + Table 2: The caption just seems like a repeat of the text in the section. Please consider revising this.

We shortened the table caption.

2.15  + Line #207: Here you mention that a grid spacing of 40 km will be used for the nested simulation, but there is still no justification as to why this resolution was chosen and why it is thought to be appropriate. See other comments above.

We included this in the introduction.

2.16  + Line #228: Please revise the sentence: '...on the one hand and exclude tropospheric clouds on the other hand'.

We revised the sentence and added Pitts et al. (2018) as citation.

2.17  + Lines #252-255: This is the sort of information that needs to have been included much earlier, say at the end of the Introduction. So that you are clearly explaining early on to the reader the scope of the paper.

We included this in the introduction.

2.18  + Line #259: I think that the results are much more nuanced and subtle than simply saying 'can be directly simulated with the resolution of 40 km'. Especially, because you are not showing any evidence at this point that the 40 km simulation of the wave is realistic.

We deleted this sentence since it is discussed better in the later paragraphs of this section.

2.19  + Lines #262-264: Please revise this text. As already mentioned above, there is more to accurately representing a mountain wave than just resolving the steepness/height of the orography. This is a rather naive understanding of the problem.

We revised the whole paragraph.

2.20  + Figure 4: The 80 km results don't seem to be mentioned.

We mentioned it as the transition between these two resolutions.

2.21 + Line #265: Its not clear what you are referring to by 'Therefore, the flow over the mountain range is under represented in the model'. The global model with a much smoother/lower orography might actually have enhanced flow over the Peninsula, although this is not a given as the model has a sub-grid scale orographic drag scheme to represent unresolved drag. Please revise.

We deleted the sentence.

2.22 +Lines #265-266: This is not clear. Please revise.

We have rephrased the sentences.

2.23 + Line #275: As I have tried to emphasize, I think that you need to make these sort of statements more comparative, ie compared to the 160 km model, the 40 km model represents a well defined mountain wave with an amplitude of 10 K – this wave is entirely absent in the 160 km model. So you are trying to justify that you have produced a step change improvement in the representation of mountain waves in the model by going from 160 to 40 km.

We adapted this sentence and all the sentences where such statements occurred.

2.24 + Line #280: Spelling. Fourth.

Corrected.

2.25  + Line #314: Not clear what 'datasets' you are referring to here. Please clarify in the text.

We replaced it by "CALIOP and ICON-ART".

2.26  + Line #317: Its not at all clear what 'the results can be found in Fig. 5' are referring to. Maybe you are referring to the model v CALIOP PSC volume concentration, but this was introduced many paragraphs previously.

We replaced it by "comparison between CALIOP and ICON-ART".

2.27  + Figure 5: I'm not convinced that a statistical analysis is possible for such a short period considered. Please justify that this in the text, or amend the language so that you say that you are simply making a comparison for the period examined. For example, what is the frequency of CALIOP measurements etc.

We added the number of grid points that are in the nest during the mountain wave event to the CALIOP discussion. In total, we get more than 0.5 million grid points with PSCs in CALIOP and in ICON-ART.

2.28 + Figure 5: Its not clear what the temperature values are on the figure. The ICON-ART panel has two different sets of values, which are unclear. The CALIOP panel has different values, which range from 23K to 349K (I think this is wrongly labelled). This is confusing, and makes it unclear how to compare the model results v CALIOP. Consequently, I found it difficult to follow the explanation of the Figure 5 in lines #324-330.

As mentioned in the figure caption, the numbers on the right hand side are the number of grid points with PSCs. To make it clearer, we added this to the panels and also added an example in the figure caption of Fig. 5.

2.29 + Lines #324-330: You haven't referred to where the improvement resulting from resolving mountain waves would be expected in terms of PSCs. Presumably, you would expect more of a benefit for ice PSCs due to their lower temperature formation threshold compared to NAT and STS PSCs, ie they require the additionally cooling from GWs to exceed this temperature. Please clarify this when referring to Figure 5.

This can be seen quite nicely by the new panels added in response to your last Major Comment. Discussion of this point is included, too.

2.30 + Lines #367-369: How important are these missing fine scale features? What are the implications for the simulated GW temperature perturbations? Does this suggest that ideally a higher resolution is required? Can you connect these deficiencies to the results in Figure 5, 6 and 7?

We included some statements about the fine structures, in Sect. 5.2 and 5.3, which also connect them better.

2.31    + Figure 11: The 80 km results are included but not discussed. How do these compare with the 40 km results?

We included a sentence to ice PSCs and the 80 km resolution.

2.32    + Figures 13: It's a little confusing jumping from the Peninsula region to a global domain. Perhaps this needs a separate subsection or added to the section describing Figure 14.

We moved the section describing Fig. 14 upwards.

---

## Referee Report (RR1)

The authors have obviously taken on board all of the reviewer's suggestions, and done their utmost to address them. The manuscript is much improved as a result, and crucially more readable and accessible. This is an excellent paper, with some really novel results that have been carefully analysed. I particularly thought that the section on the influence of gravity waves on PSC precursors was interesting and novel. However, there are still some minor comments that I have listed below, which should be straightforward to address given the competence of the team, but nevertheless important. I particularly feel that much of these changes are necessary to increase the authority of the paper.

More importantly, there is maybe a more major issue of the use of reanalysis being used to semi-validate the nested simulations in section 5.1, which is misplaced and undermines section 5.3 - surely a more coherent order would be to combine these two sections and/or just focus on the use of AIRS and keep the reanalysis data solely for describing the synoptic conditions responsible for the wave event. I'm sure that would result in a much cleaner and stronger paper, and a very strong argument for both the dynamical and PSC chemistry benefits of the nested ICON-ART configuration.

Once these comments are addressed I recommend that the manuscript is accepted.

Major/minor comments

- Figure 5: You have mentioned that reanalysis resolves the gravity wave / temperature amplitude (e.g. your Figure 4). If you wanted to properly test this (and I would have thought show conclusively that it only partially resolves this, at least when compared to the 40 km model), then you could include a column in Figure 5 that shows results based on reanalysis. Or perhaps, this could be included as a supplementary figure. I have to say that the paper does give a rather confusing message as to what are the limitations of reanalysis / coarse resolution (of order ~100 km), so this would go a long way to clarifying this.

  However, if you would rather not do this, then please include clear evidence in the manuscript in the form of citations that details the ability of modern reanalysis to capture small-scale orographic gravity waves. Four other comments related to this issue are also on lines #195, #230, #243, and #375 as well as Section 5.3. Even better, as I explain below, it would surely be much better to just stick to comparing the model against AIRS, which shows great promise. This is honestly all that you need, and any other argument just gets confusing and convoluted.

- Line #195: There seems to some confusion here. You wisely have included AIRS to validate the model representation of the gravity wave event (as done elsewhere), particularly I assume the temperature perturbation induced by the wave. Therefore, why is there considerable mention of the model being compared to the reanalysis representation of the wave event? Surely this is flawed and/or redundant?
- Line #230: Here you are explaining that (relatively coarse resolution, resolution ~80 km) ERAI is able to resolve the temperature minimum induced by gravity waves to the lee of the Peninsula. How can a grid spacing of 80 km resolve a mountain wave of around 300 km wave length? Especially as your Figure 5 shows a marked difference between 80 and 40 km results. Please modify/soften the language so that this remark is less glaring. Maybe comment that there is evidence of a wave in ERAI, but given the resolution it is likely poorly

resolved, amplitude underestimated, etc…. See Alexander and Teitelbaum (2007), which you refer to.  See also comment above.

- Line #243: Why would you expect temperatures to be in agreement with ERA-Interim? As mentioned above, your 40 v 80 km results are not in agreement, so why should ERAI v 40 km results be in agreement?  Perhaps the agreement is because the location is over the base of the Peninsula, so conducive to more broader horizontal scale waves that reanalysis products can resolve, compared to waves forced by the much narrower Peninsula region further north (width ~100km). My understanding is that it is well established that reanalysis products fail to capture the detailed temperature structure associated with gravity waves, which is why you have included AIRS data in your analysis. Maybe Alexander and Teitelbaum (2007) is appropriate, as it states that the fine details apparent in AIRS is not evident in ECMWF data.  See also comment above.
- Section 5.3: My initial reaction to section 5.3 is that is a repeat of 5.1, and why are two separate sub-sections required?  Surely a better place for the AIRS comparison would be 5.1, so that the dynamics / simulation of the mountain wave is dealt with in one place. This would also be a lot cleaner and enable you to easily remove all comparisons/mention of agreement with reanalysis that are currently in 5.1.  In any case, how can Sect. 5.1 improve on 5.3?  Surely a state of the art comparison with AIRS in 5.3 makes the results of 5.1 largely redundant, and in doing so lessens the impact of 5.3?
- Line #375: Here you categorically state that the model matches AIRS. Excellent result, which negates the need for any earlier and confusing mentioning of the good agreement with reanalysis.

Minor comments/changes

- Line #2: Spelling. Surfaces.
- Line #4: Please mention the possible role of unresolved non-orographic waves somewhere in the manuscript, and how the ICON-ART model would possibly result in an improvement in this aspect.  Please see Tritscher et al. (2021) for a (short) review/discussion of the possible role of non-orographic waves and PSCs.
- Line #29: This statement is not quite true. The requirement is that a sizeable component of the wind is perpendicular to the barrier, not all of the large scale flow. This is why south-easterly winds, which are common over the Peninsula, are closely associated with forcing orographic gravity waves.
- Line #60: Please revise sentence beginning 'Thus, a low resolution ….'. Its currently unclear.
- Line #75: Please revise use of 'large' here. Horizontal wavelengths of ~300 km are not large, but mesoscale.
- Line #183: The list of PSC types given here could do with a little more detail/explanation, rather than expecting the reader to work this out for themselves via Pitts et al. 2018. For example, what are enhanced NAT mixtures, Wave-ice, etc? How reliable are CALIOP measurements of the different PSC types?
- Figure 4 caption: What does 'dark shadow' refer to? It made me think of the Lord of the Rings!! I think this is a translation issue from German to English. Please correct.
- Line #239: I mentioned this in my first review. Please be careful how you word this. The interaction between the flow and the detailed orography is better captured at higher resolution, but you are saying something different (flow over the mountain in the Antarctic Peninsula nest is improved). Also, even if the model orography was to converge at higher

resolution (ie differences in orography between the different resolutions becomes small), you might still expect differences in the representation of the key features of the orographic gravity wave due to differences in the grid spacing / resolving finer details of the dynamics. Such features are reviewed by Smith et al. 1989.

Smith, R. B. (1989). Hydrostatic airflow over mountains. *Advances in Geophysics*, 31, 1-41, https://doi.org/10.1016/S0065-2687(08)60052-7.

- Line #248: This argument is rather confused here. Is it the 40 km horizonal resolution or the 500 m vertical resolution that are important, or both. Please clarify.
- Line #257: Include a reference here after 'consistent with theory'. Smith (1989), mentioned above, is often acknowledged as being the classic paper for mountain waves.
- Line #284: Have these parameters (particularly the subscripts) been previously defined? What does the subscript 'NAT|ice' refer to? Please make sure that all parameters are clearly defined.
- Figure 6: I think it is worth commenting that in both the 40 km model and CALIOP that the fraction of Wave-ice PSCs is very small. Is this to be expected? How reliable are CALIOP measurements of Wave-ice PSCs?
- Line #321: I don't think that you can say with any certainty that a higher resolution is required to get the number of ice PSCs correct. Also, its not clear what would be the impact of higher resolution? Are you suggesting that the temperature perturbation amplitude would be larger at higher resolution, and hence more realistic? If so, then please make this clear. You also made a similar comment at line #339 and #348, so please amend this also.
- Line #330: You have identified a deficiency in the simulation of STS. But no explanation is given as to why this is occurring, or indication that this will be looked at later. Please amend this.
- Line #364: Again, you can not state that a higher resolution would resolve this issue. You don't know this for certain and its beyond the scope of your work to examine this. You need to soften this language, so say that it is perhaps probable that this is a resolution issue (or consistent with a resolution dependence issue), which would perhaps be corrected by going to higher grid spacings, as shown by ……
- Line #367: Please refer to Figures 5, 6 and 7 of Orr et al. 2020, which examine the impact of a higher resolution simulation (via a parameterisation) on the simulated temperatures relative to the ice temperature threshold, as well as the formation potential of ice PSCs.
- Line #370: I would suggest prefacing the reference to Figure 9 with something like 'Following Orr et al. (2015) …', as this is the same model v AIRS comparison as that study undertook.
- Line #372: You could perhaps mention that an analogous study by Orr et al. (2015) using a 4 km model seems to suggest a better match between observed v simulated wave amplitudes over the Antarctic Peninsula, ie more realistic and larger at the higher resolution. This is perhaps the evidence that you need to suggest that a further extension of your study could be to perhaps go to higher grid spacings.
- Line #404: Meteorological models do not have 'ice schemes'. Not sure what you are meaning here, so please revise.
- Figure 14: Perhaps worth comparing this to the results of Figure 10 in Orr et al. (2020), which also looked at the effects on chlorine activation. Similarly, Figure 15 could also be compared to Fig 11 in Orr et al., which also looked at the impacts on ozone.
- Conclusions/outlook: Maybe a mention of the importance of unresolved non-orographic gravity waves could be included, e.g. from SH storm tracks.

---

## Author Response (AR2)

**Response to Referee #2**

Dear referee,

thank you again for this detailed review of our manuscript and your comments. We would like to emphasise that it wasn't our intention to use ERA-Interim to validate our simulations and corrected all statements in the manuscript with respect to this. It should rather show the large-scale meteorological conditions during the mountain wave event.

Major changes include:

- replacing the temperature in Fig. 4 by the near-surface wind speed

- moving the AIRS comparison between Fig. 4 and 5 in Sect. 5.1

Yours sincerely and on behalf of all co-authors,
Michael Weimer

**1 Major comments**

**1.1 - Figure 5: You have mentioned that reanalysis resolves the gravity wave / temperature amplitude (e.g. your Figure 4). If you wanted to properly test this (and I would have thought show conclusively that it only partially resolves this, at least when compared to the 40 km model), then you could include a column in Figure 5 that shows results based on reanalysis. Or perhaps, this could be included as a supplementary figure. I have to say that the paper does give a rather confusing message as to what are the limitations of reanalysis / coarse resolution (of order ~100 km), so this would go a long way to clarifying this. However, if you would rather not do this, then please include clear evidence in the manuscript in the form of citations that details the ability of modern reanalysis to capture small-scale orographic gravity waves. Four other comments related to this issue are also on lines #195, #230, #243, and #375 as well as Section 5.3. Even better, as I explain below, it would surely be much better to just stick to comparing the model against AIRS, which shows great promise. This is honestly all that you need, and any other argument just gets confusing and convoluted.**

As stated above, it wasn't our intention to validate our simulations with ERA-Interim. We want to show the meteorological conditions that led to the mountain wave event. Therefore, we replaced the 25 km temperature by the 10 m wind speed which is now colour-coded in Fig. 4. Additionally, we removed every statement comparing ICON-ART with ERA-Interim from the manuscript.

**1.2  - Line #195: There seems to some confusion here. You wisely have included AIRS to validate the model representation of the gravity wave event (as done elsewhere), particularly I assume the temperature perturbation induced by the wave. Therefore, why is there considerable mention of the model being compared to the reanalysis representation of the wave event? Surely this is flawed and/or redundant?**

We hope that our explanation to Comment 1.1 and the changes made in the manuscript removed this confusion.

**1.3  - Line #230: Here you are explaining that (relatively coarse resolution, resolution ~80 km) ERAI is able to resolve the temperature minimum induced by gravity waves to the lee of the Peninsula. How can a grid spacing of 80 km resolve a mountain wave of around 300 km wave length? Especially as your Figure 5 shows a marked difference between 80 and 40 km results. Please modify/soften the language so that this remark is less glaring. Maybe comment that there is evidence of a wave in ERAI, but given the resolution it is likely poorly resolved, amplitude underestimated, etc.... See Alexander and Teitelbaum (2007), which you refer to. See also comment above.**

As mentioned above, we want to show the meteorological conditions and not validate our simulations with ERA-Interim.

**1.4**    **- Line #243: Why would you expect temperatures to be in agreement with ERA-Interim? As mentioned above, your 40 v 80 km results are not in agreement, so why should ERAI v 40 km results be in agreement? Perhaps the agreement is because the location is over the base of the Peninsula, so conducive to more broader horizonal scale waves that reanalysis products can resolve, compared to waves forced by the much narrower Peninsula region further north (width ∼100km). My understanding is that it is well established that reanalysis products fail to capture the detailed temperature structure associated with gravity waves, which is why you have included AIRS data in your analysis. Maybe Alexander and Teitelbaum (2007) is appropriate, as it states that the fine details apparent in AIRS is not evident in ECMWF data. See also comment above.**

We removed this statement.

**1.5**    **- Section 5.3: My initial reaction to section 5.3 is that is a repeat of 5.1, and why are two separate sub-sections required? Surely a better place for the AIRS comparison would be 5.1, so that the dynamics / simulation of the mountain wave is dealt with in one place. This would also be a lot cleaner and enable you to easily remove all comparisons/mention of agreement with reanalysis that are currently in 5.1. In any case, how can Sect. 5.1 improve on 5.3? Surely a state of the art comparison with AIRS in 5.3 makes the results of 5.1 largely redundant, and in doing so lessens the impact of 5.3?**

We followed your suggestion and moved the AIRS comparison to Sect. 5.1. With this, the evaluation with measurements always precedes the model domain comparisons. In addition, the section with the CALIOP comparison can build on the AIRS results which could be a better indicator that a higher resulution could improve the results.

**1.6  - Line #375: Here you categorically state that the model matches AIRS. Excellent result, which negates the need for any earlier and confusing mentioning of the good agreement with reanalysis.**

We agree with this statement and moved the AIRS comparison to Sect. 5.1.

**2  Minor comments/changes**

**2.1  - Line #2: Spelling. Surfaces.**

Corrected to "surface".

**2.2  - Line #4: Please mention the possible role of unresolved non-orographic waves somewhere in the manuscript, and how the ICON-ART model would possibly result in an improvement in this aspect. Please see Tritscher et al. (2021) for a (short) review/discussion of the possible role of non-orographic waves and PSCs.**

We added a sentence at the very end of the conclusions that they are not considered in our simulations.

**2.3  - Line #29: This statement is not quite true. The requirement is that a sizeable component of the wind is perpendicular to the barrier, not all of the large scale flow. This is why south- easterly winds, which are common over the Peninsula, are closely associated with forcing orographic gravity waves.**

We added this to the statement.

**2.4  - Line #60: Please revise sentence beginning "Thus, a low resolution ....". Its currently unclear.**

We revised the sentence to "Thus, a global low-resolution simulation provides boundary conditions for a region with refined grid, similar to mesoscale models."

**2.5   - Line #75: Please revise use of "large" here. Horizontal wavelengths of 300 km are not large, but mesoscale.**

We replaced "large" by "mesoscale".

**2.6   - Line #183: The list of PSC types given here could do with a little more detail/explanation, rather than expecting the reader to work this out for themselves via Pitts et al. 2018. For example, what are enhanced NAT mixtures, Wave-ice, etc? How reliable are CALIOP measurements of the different PSC types?**

We included some statements explaining the categories and their reliability based on Pitts et al. (2018).

**2.7   - Figure 4 caption: What does "dark shadow" refer to? It made me think of the Lord of the Rings!! I think this is a translation issue from German to English. Please correct.**

We obviously don't want to be Sauron or one of his followers and corrected the figure caption.

**2.8   - Line #239: I mentioned this in my first review. Please be careful how you word this. The interaction between the flow and the detailed orography is better captured at higher resolution, but you are saying something different (flow over the mountain in the Antarctic Peninsula nest is improved). Also, even if the model orography was to converge at higher resolution (ie differences in orography between the different resolutions becomes small), you might still expect differences in the representation of the key features of the orographic gravity wave due to differences in the grid spacing / resolving finer details of the dynamics. Such features are reviewed by Smith et al. 1989. Smith, R. B. (1989). Hydrostatic airflow over mountains. Advances in Geophysics, 31, 1-41, https://doi.org/10.1016/S0065-2687(08)60052-7.**

We corrected the statement.

**2.9 - Line #248: This argument is rather confused here. Is it the 40 km horizonal resolution or the 500 m vertical resolution that are important, or both. Please clarify.**

We clarified the statement. It's both horizontal and vertical resolution that is important.

**2.10 - Line #257: Include a reference here after "consistent with theory". Smith (1989), mentioned above, is often acknowledged as being the classic paper for mountain waves.**

We already mentioned Queney (1947) as reference some lines above, but added Smith (1989), too, now at both lines, so that it is clear what is our reference.

**2.11 - Line #284: Have these parameters (particularly the subscripts) been previously defined? What does the subscript "NAT|ice" refer to? Please make sure that all parameters are clearly defined.**

Yes, they are defined in Sect. 4.1 with the CALIOP description. We added a reference to this section when mentioning these thresholds again.

**2.12 - Figure 6: I think it is worth commenting that in both the 40 km model and CALIOP that the fraction of Wave-ice PSCs is very small. Is this to be expected? How reliable are CALIOP measurements of Wave-ice PSCs?**

We moved the discussion of the Wave-ice category, which was at the end of the section, up to the discussion of Fig. 6 (now Fig. 8) and added a sentence about the reliability of the Wave-ice category, based on the statements above.

**2.13**  - Line #321: I don't think that you can say with any certainty that a higher resolution is required to get the number of ice PSCs correct. Also, its not clear what would be the impact of higher resolution? Are you suggesting that the temperature perturbation amplitude would be larger at higher resolution, and hence more realistic? If so, then please make this clear. You also made a similar comment at line #339 and #348, so please amend this also.

We amended all three statements and included the reference to Orr et al. (2020) as suggested some comments below.

**2.14**  - Line #330: You have identified a deficiency in the simulation of STS. But no explanation is given as to why this is occurring, or indication that this will be looked at later. Please amend this.

We added a sentence that it should be analysed in future simulations.

**2.15**  - Line #364: Again, you can not state that a higher resolution would resolve this issue. You don't know this for certain and its beyond the scope of your work to examine this. You need to soften this language, so say that it is perhaps probable that this is a resolution issue (or consistent with a resolution dependence issue), which would perhaps be corrected by going to higher grid spacings, as shown by ...

We corrected the statement accordingly.

**2.16**  - Line #367: Please refer to Figures 5, 6 and 7 of Orr et al. 2020, which examine the impact of a higher resolution simulation (via a parameterisation) on the simulated temperatures relative to the ice temperature threshold, as well as the formation potential of ice PSCs.

We included the reference.

**2.17**   **- Line #370: I would suggest prefacing the reference to Figure 9 with something like "Following Orr et al. (2015) ...", as this is the same model v AIRS comparison as that study undertook.**

We added a sentence at the beginning of this section (originally line 354) saying that the same methodology was also applied in Orr et al. (2015).

**2.18**   **- Line #372: You could perhaps mention that an analogous study by Orr et al. (2015) using a 4 km model seems to suggest a better match between observed v simulated wave amplitudes over the Antarctic Peninsula, ie more realistic and larger at the higher resolution. This is perhaps the evidence that you need to suggest that a further extension of your study could be to perhaps go to higher grid spacings.**

We added this to the paragraph.

**2.19**   **- Line #404: Meteorological models do not have "ice schemes". Not sure what you are meaning here, so please revise.**

We replaced it by "hydrometeor microphysics".

**2.20**   **- Figure 14: Perhaps worth comparing this to the results of Figure 10 in Orr et al. (2020), which also looked at the effects on chlorine activation. Similarly, Figure 15 could also be compared to Fig 11 in Orr et al., which also looked at the impacts on ozone.**

Since the figure with NAT differences in Orr et al. (2020) shows slightly different things (temporal variability and surface concentration in contrast to our analysis) we added a qualitative statement comparing the altitude regions for late July. We included a sentence comparing the ozone result with Orr et al. (2020).

**2.21    - Conclusions/outlook: Maybe a mention of the importance of unresolved non-orographic gravity waves could be included, e.g. from SH storm tracks.**

We added this at the end of the conclusions.